# Twenty years of tuberculosis-driven selection shaped the evolution of the meerkat major histocompatibility complex

Nadine Müller-Klein ●[1] ✉, Alice Risely ●[1,2], Kerstin Wilhelm ●[1], Vanessa Riegel ●[1], Marta Manser ●[3,4,5], Tim Clutton-Brock ●[4,5,6], Luke Silver ●[7,8], Pablo S. C. Santos ●[1], Dominik W. Melville ●[1,9] & Simone Sommer ●[1,9]

Pathogen-mediated balancing selection (PMBS) drives host evolution across the tree of life. Distinguishing between the three main mechanisms underlying PMBS, that is, rare-allele advantage, fluctuating selection and heterozygote advantage, remains difficult, limiting our understanding of frequency-dependent adaptations by hosts and counter-adaptation by pathogens. Here we leverage immune genetic and disease surveillance data from over 1,500 wild meerkats (*Suricata suricatta*) to track how selection by the tuberculosis (TB)-causing *Mycobacterium suricattae* shaped the evolution of the meerkats' major histocompatibility complex (MHC) over two decades. Compared with neutral genetic markers, we detect more rapid differentiation and recycling of alleles at the MHC-DRB loci, suggesting that TB imposes strong PMBS on wild meerkats. In addition, we show that meerkats carrying the MHC allele Susu-DRB*13 were initially more likely to develop clinical signs of TB, with the effect reversing over the course of the study, followed by an increase in the frequency of Susu-DRB*13. Meerkats carrying Susu-DRB*13 also showed slower progression to TB signs and longer survival once signs of TB manifested. Lifetime reproductive success reflected the resilience effect conferred by Susu-DRB*13. Based on several lines of evidence, we propose that rare-allele advantage or fluctuating selection, rather than heterozygote advantage, drive our observation in this longitudinally sampled wild mammal population.

Balancing selection maintains much of the biological diversity on our planet[1]. Pathogen-mediated balancing selection (PMBS), in particular, shapes the exceptionally high polymorphism commonly observed in genomic regions encoding innate[2,3] and adaptive[4,5] immune responses. In vertebrates, the major histocompatibility complex (MHC) is the most polymorphic and best-understood genetic basis of pathogen resistance[6,7]. MHC molecules are an essential cell surface component of the immunological cascade that recognizes and presents pathogen-derived peptides to T cells[8]. A central tenet of PMBS in

maintaining MHC polymorphism is the rare-allele advantage, which emerges from pathogens evolving to evade common MHC variants. Thus, rare variants, if advantageous, increase in frequency until pathogens counter-adapt[6,9,10]. This sets up a cyclic arms race between hosts and pathogens that theoretically maintains high levels of MHC polymorphism[11]. Yet, such negative frequency-dependent selection on the MHC has been documented only in experiments on short-lived model organisms, such as mice and fish[12–14], or inferred from snapshots in time[15–17]. Robust evidence from natural populations documenting the

entire cycle in which a common MHC allele becomes rare following pathogen evasion and then regains resistance is lacking.

One explanation for the lack of evidence for the rare-allele advantage, especially from natural systems, is that findings are often also compatible with other mechanisms of PMBS, such as heterozygote advantage[9,18] and fluctuating selection[19,20]. The heterozygote advantage hypothesis is conceptually related to the rare-allele advantage in that both stem from the idea that single alleles can bind peptides from a limited range of pathogens. Hosts with a greater number or a more divergent repertoire of alleles are therefore expected to mount efficient responses against a wider range of pathogens[4,21–23]. Fluctuating selection arises from spatio-temporal variation in PMBS, resulting in the enormous allelic diversity seen at the species level, while maintaining a more limited subset in local populations[24,25]. Owing to their conceptual similarities, host–pathogen associations may resemble any of these mechanisms, making a clear differentiation difficult[6,9,26,27]. However, depending on the underlying mechanisms, the MHC is expected to evolve differently compared with neutral markers[9,25,27–29]. Hence, delineating between mechanisms requires a comparison of MHC and neutral genetic evolution, and a contrast of MHC–pathogen associations over space and time, ideally across multiple populations[6,9].

Although bacteria of the *Mycobacterium tuberculosis* complex cause debilitating, often lethal, tuberculosis (TB) infections in humans[30] and wildlife[31], evidence for long-term selection is lacking[32–35]. Here we leverage two decades of *Mycobacterium suricattae* surveillance, individual disease records and new MHC class II data from over 1,500 wild meerkats (*Suricata suricatta*) living in tight social groups at the Kuruman River Reserve in the Kalahari Desert, South Africa. In the cooperative breeding system of meerkats, a dominant breeding pair largely monopolizes reproduction and employs group members to help in pup care[36]. The pair asserts their dominance through aggression within their social group[37], and aggressive feuds over territories between neighbouring groups are common[38]. Aggression is also presumed to be among the main behavioural mechanisms by which the meerkat-specific *M. suricattae*, first reported in the 1990s, spread through the population[39–41]. After a long latent period, the infection causes swelling of the submandibular, inguinal and cervical lymph nodes, progressing to lesions, physical deterioration (Fig. 1a) and death within an average of 6.6 months after clinical signs appear[39,42,43]. While TB curbs survival, disease progression varies widely, with some individuals surviving up to 7 years after the onset of clinical signs[39]. This variability provides a unique opportunity to investigate the links between TB, immunogenetics and fitness in a longitudinally monitored wildlife population (Supplementary Fig. 1 and Supplementary Table 1). First, we describe MHC class II diversity in meerkats and then test whether the evolution at the MHC outpaces evolution at neutral genetic markers, indicating that PMBS affects host immunogenetics rather than microsatellite markers. Second, we hypothesize that the MHC composition and functional diversity explains interindividual differences in meerkat TB susceptibility, survival and lifetime reproductive success better than neutral genetic diversity and compare them with other biological, socio-ecological and environmental covariates. Finally, we attempt to distinguish between three mechanisms of PMBS, that is, rare-allele advantage, heterozygote advantage and fluctuating selection (Fig. 1b). Indeed, we document that a single allele correlated with TB susceptibility, survival and lifetime reproductive success, and we draw from several lines of evidence to suggest that the dominant mechanisms are likely to be rare-allele advantage or fluctuating selection.

## Results

### Meerkat MHC class II DRB polymorphism and neutral genetic evolution

High-throughput sequencing identified 37 functional MHC-DRB exon 2 alleles in 1,567 meerkats (Fig. 2a,b). Meerkats carried between

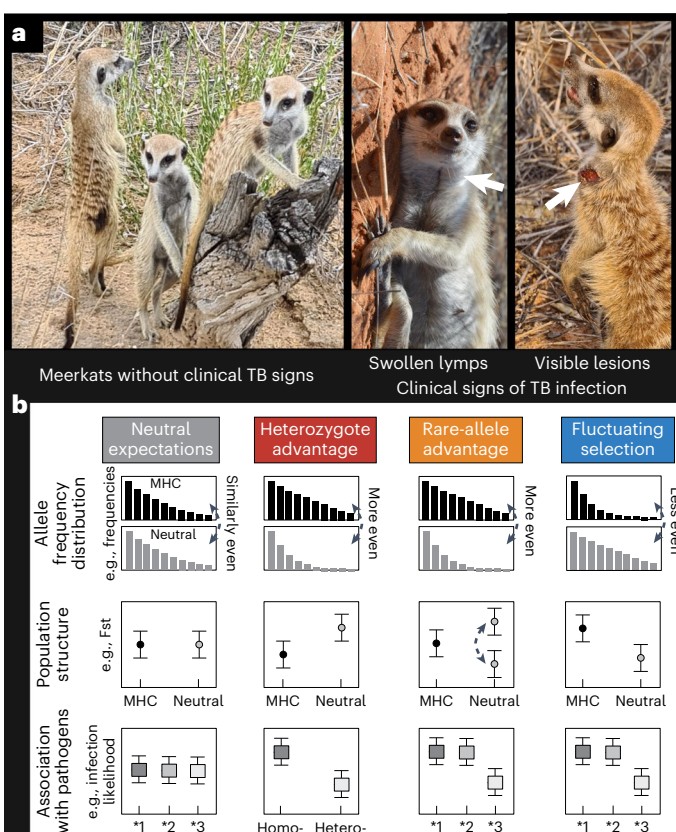

**Fig. 1 | Indicators for meerkat TB infection and alternative mechanisms of PMBS. a**, Contrast between meerkats without clinical signs of *Mycobacterium suricattae* infection and those displaying swelling of head lymph nodes (for example, white arrow in central picture), open lesions (for example, white arrow in right picture) and physical deterioration. **b**, A comparison between allele frequency distribution, population structure and MHC–pathogen association allows to differentiate evolution at the MHC from evolution at neutral markers and to inferentially distinguish between possible non-mutually exclusive mechanisms of PMBS (visually adapted from ref. 9). Meerkat photo credit: Simone Sommer and Shaylee Alderton. This figure was created with BioRender.com.

one and seven MHC alleles (median: 2, mean: 2.5 ± 1.1 standard deviation (s.d.)), suggesting the presence of at least four distinct DRB loci. This was confirmed by manual annotation of the MHC class II genes, where we identified two complete DRB loci and an additional two exon 2 fragments as well as two potential pseudogenes (Supplementary Tables 4 and 5). All functional alleles map to exon 2 of the functional DRB loci (for more details, see Supplementary Results).

MHC alleles varied in their relative abundance (referred to as frequency henceforth) over time (Fig. 2c), with average individual expected microsatellite heterozygosity ($H_{exp}$) remaining stable over time. We found that MHC allele frequencies were more evenly distributed compared with those of neutral genetic markers (mean difference: 0.08; paired $t$-test −8.10, $P < 0.001$; Fig. 2d), possibly arguing for recycling of MHC alleles in line with a frequency-dependent rare-allele advantage. Besides, MHC and microsatellite fixation index (Fst) diverged over time (Mantel test on microsatellites: $P = 0.001$; slope $1.1 × 10^{-3}$, confidence interval (CI) $1.0 × 10^{-3}$ to $1.2 × 10^{-3}$; on MHC markers: $P = 0.001$; slope $1.9 × 10^{-3}$, CI $1.7 × 10^{-3}$ to $2.0 × 10^{-3}$). The molecular divergence was higher at the MHC-DRB alleles than estimated for the microsatellites (interaction: $P < 0.001$; estimate $−7.8 × 10^{-4}$; CI $8.9 × 10^{-4}$ to $6.6 × 10^{-4}$; Fig. 2e and Supplementary Table 6), suggesting stronger selection acting on MHC alleles.

The 37 functional alleles clustered into 12 functionally distinct MHC supertypes (Supplementary Table 7 and Supplementary Fig. 2a)

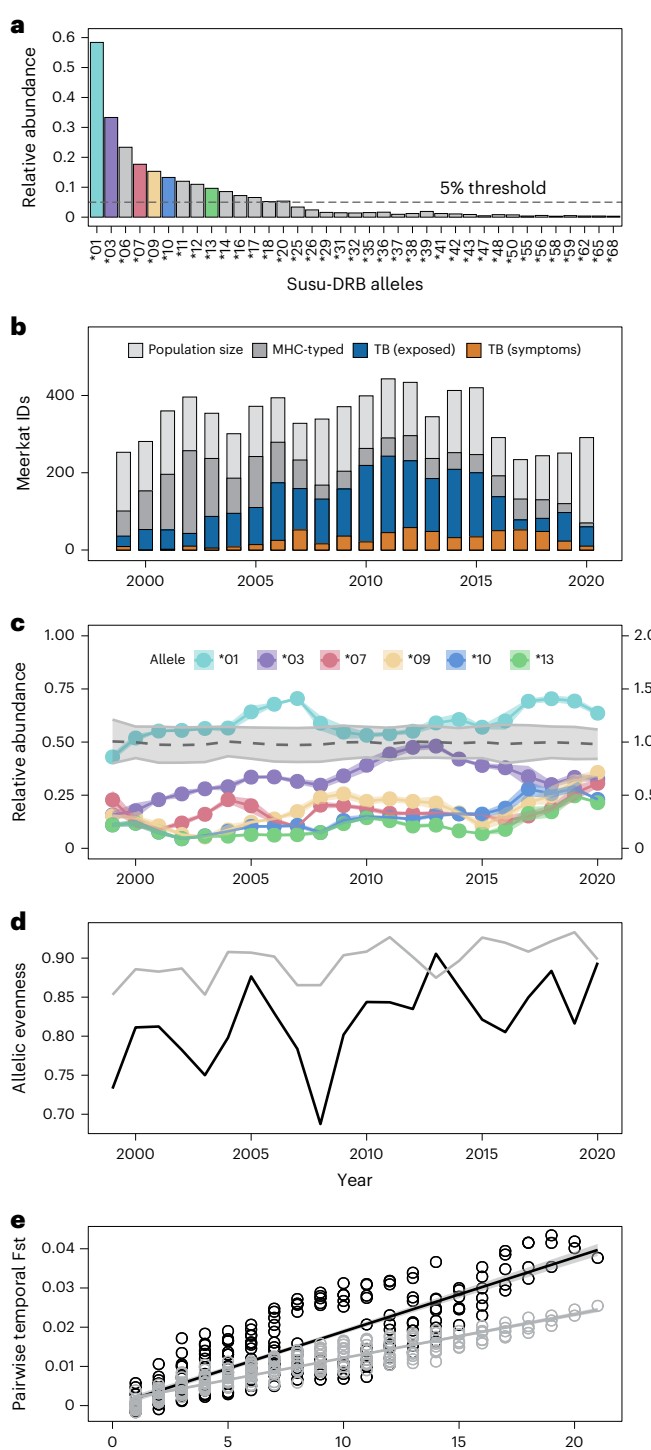

**Fig. 2 | Meerkat MHC class II diversity and neutral genetic evolution over time. a**, The relative abundance of MHC-DRB alleles among 1,567 sampled meerkats. The dashed line indicates the 5% threshold. Coloured boxes represent alleles with data available for all study years. **b**, The total number of uninfected (dark grey), TB-exposed (blue) and TB-positive (orange) individuals with MHC genotype information compared with the rest of uncharacterized meerkats in the population (light grey). **c**, Change in relative abundance (termed frequency; mean ± s.d.) of the six Susu-DRB alleles present in at least 5% of the population in each year since 1999. The dashed line and shaded area represent neutral genetic diversity (measured as $H_{exp}$; mean ± sd). **d**, Estimated annual evenness of MHC (black) and microsatellite (grey) allele frequencies. **e**, Fitted regression (±95% CIs) of genetic differentiation (that is, pairwise Fst) of the meerkat population over time estimated from microsatellite (grey) and MHC data (black).

and combinations of these alleles were probabilistically sorted into 16 functional haplotypes (Supplementary Table 8 and Supplementary Fig. 2b). Individuals carried between one and six supertypes (median: 2, mean 2.3 ± 0.9 s.d.). Most individuals carry two haplotypes, with only 14.6% classified as homozygotes. Any further analyses were performed on functional MHC alleles, supertypes and haplotypes occurring in more than 5% of the study population and in all years.

## MHC associations with TB and fitness

First, we identified four MHC alleles (Susu-DRB*01, Susu-DRB*03, Susu-DRB*07 and Susu-DRB*13; Fig. 3a), six supertypes (ST01, ST02, ST03, ST04, ST05 and ST09; Supplementary Fig. 3a) and one haplotype (F; Supplementary Fig. 4a) to be repeatedly associated with clinical signs of TB over the 20 years studied since 1999. For each candidate MHC allele, supertype and haplotype, we ran separate models to estimate the likelihood to ever develop clinical TB signs (TB susceptibility), the likelihood to progress from exposure to clinical signs (TB progression), persist with a clinical TB infection (TB resilience), survive and reproduce (lifetime reproductive success). All models controlled for MHC and neutral genetic diversity, as well as biological, socio-ecological, and environmental covariates, and presented *P* values are false discovery rate corrected. Because we were interested in whether pathogen–MHC associations vary over time, we included interactions between the linear and quadratic terms of year and MHC measures (see Methods for more detail).

Carrying Susu-DRB*13 influenced TB susceptibility over time (estimate −1.16, CI −2.02 to −0.31; *P* < 0.01; Fig. 3b and Supplementary Table 9). Meerkats with Susu-DRB*13 experienced a heightened risk of developing TB signs between 1999 and 2010. This pattern reversed by 2013, when individuals carrying the Susu-DRB*13 allele had a lower likelihood of developing clinical signs compared with those with other alleles (Fig. 3b). Susu-DRB*13 is the only common allele (occurrence >5%) to feature an arginine at the positively selected site (PSS) 56 and a serine at 57 (Supplementary Data 1). Similarly, individuals with haplotype F, which included Susu-DRB*13, faced higher risk until the early 2010s before the risk of developing signs declined (estimate −5.04, CI −8.57 to −1.51, *P* < 0.05, Supplementary Fig. 4b,c and Supplementary Table 9c), although this analysis relied on a smaller sample size. By contrast, TB susceptibility was generally increased in meerkats carrying Susu-DRB*07 (estimate 0.68, CI 0.25 to 1.21, *P* < 0.01; Supplementary Table 9), whereas individuals carrying alleles clustering into the supertype ST04 showed low susceptibility initially until approximately 2010 when individuals with ST04 became more likely to develop TB signs (estimate 2.38, CI 1.25 to 3.52, *P* < 0.001; Supplementary Fig. 3b,c and Supplementary Table 9b).

TB-exposed individuals carrying Susu-DRB*13 (and its corresponding ST09 and haplotype F) were less likely to progress to clinical signs of TB over the course of the study (estimate −1.23, CI −1.95 to −0.51, *P* < 0.01; Supplementary Table 10). A weaker influence on TB progression over time was also detected for Susu-DRB*07 (and its corresponding ST03, estimate −0.77, CI −1.26 to −0.28, *P* < 0.01; Supplementary Tables 10), ST01 (estimate 0.71, CI 0.19 to 1.24, *P* < 0.05; Supplementary Table 10) and ST04 (estimate 0.74, CI 0.13 to 1.36, *P* < 0.05; Supplementary Table 10). In addition, meerkats carrying Susu-DRB*13 lived for longer after showing first clinical signs than TB-positive individuals with other alleles (estimate 0.29, CI 0.13 to 0.44, *P* < 0.001; corresponding ST09: estimate 0.24, CI 0.08 to 0.40, *P* < 0.05; haplotype F: estimate 0.35, CI 0.10 to 0.61, *P* < 0.05; Fig. 4a and Supplementary Table 11). However, this effect on resilience was independent of year. The results of the survival models argue a similar point (Supplementary Table 12): while individuals with clinical TB signs showed a roughly fivefold mortality risk (estimate 1.50; CI 1.14 to 1.87, *P* < 0.001), TB-positive individuals carrying Susu-DRB*13 survived for longer (estimate −0.95; CI −1.17 to −0.17, *P* < 0.05; Fig. 4b). The same was true for individuals with haplotype F in the smaller sample

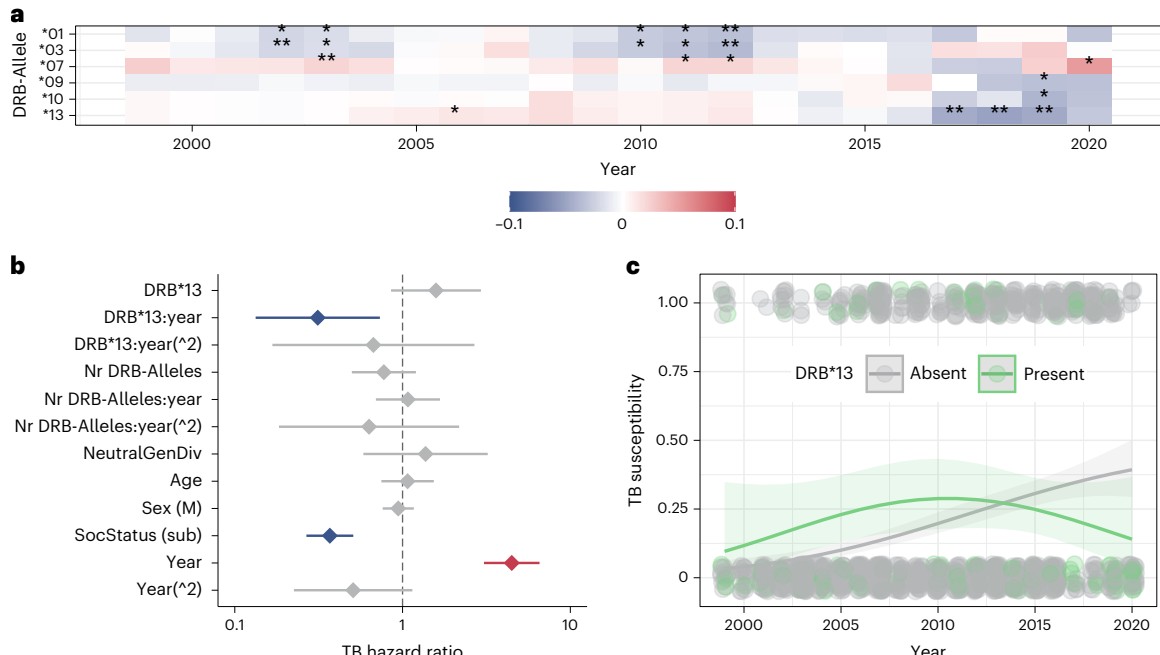

**Fig. 3 | MHC-DRB allele association with TB susceptibility. a**, A heat map displaying higher (that is, positive association) and lower (that is, negative association) likelihood of observing TB signs in meerkats with different MHC-DRB alleles based on co-occurrence analysis (*$P < 0.05$, **$P < 0.01$). **b,c**, Effect sizes (±95% CIs; $n = 1,497$) for all explanatory variables included in the generalized linear mixed-effects model including the MHC allele Susu-DRB*13 (**b**) and visualization of nonlinear effects of Susu-DRB*13 (green) on TB susceptibility over time (**c**). Significant effects are in blue (lower likelihood of developing TB signs) and red (higher likelihood of developing TB signs).

population (estimate −2.33; CI −3.91 to −0.74, $P < 0.01$; Supplementary Table 12). Finally, individual lifetime reproductive success, as the ultimate Darwinian fitness measure, was generally higher in individuals that carried Susu-DRB*13 (estimate 0.99, CI 0.23 to 1.75, $P < 0.05$) and peaked between 2005 and 2013 (estimate −2.58, CI −4.59 to −0.58, $P < 0.05$; Fig. 4c and Supplementary Table 13). By contrast, for individuals with the most common MHC allele Susu-DRB*01 or its corresponding supertype, lifetime reproductive success declined linearly over the years (Susu-DRB*01: estimate −0.85, CI −1.45 to −0.25, $P < 0.05$; ST01: estimate −0.89, CI −1.54 to −0.24, $P < 0.05$; Supplementary Fig. 5).

In addition, socio-ecological and environmental factors correlated with TB risk and fitness in meerkats: For example, dominant meerkats were more susceptible but subordinates with TB have a sevenfold lower survival chance. Higher mean temperature increased the likelihood to progress to clinical signs of TB, while reduced rainfall increased mortality risk (see Supplementary Results for detailed information; Supplementary Tables 9–13).

### Delineating between mechanisms of PMBS

Comparing between the evolution of MHC and neutral genetic markers, and looking at MHC–pathogen associations, can help differentiate among rare-allele advantage, fluctuating selection and heterozygote advantage (Fig. 1b). Yet, without replicated meerkat populations, we are left with inference and deduction. The higher evenness of MHC allele frequencies compared with those of neutral markers supports the rare-allele and heterozygote advantages, whereas the greater temporal divergence at the MHC is characteristic of rare-allele advantage or fluctuating selection. Among these mechanisms, we think heterozygote advantage is least likely because, even though in the TB susceptibility null model (that is, without specific MHC alleles) allelic diversity had a significant effect (estimate −1.56, CI −2.63 to −0.50, $P < 0.05$; Fig. 5a and Supplementary Table 9), when Susu-DRB*13 was included, this effect was not recovered (estimate −0.46, CI −1.70 to 0.78, $P = 0.466$; Supplementary Table 9). We hypothesized, therefore, that

the effect of MHC diversity was driven by the effect of this single allele. If this were the case, we would expect MHC diversity to have no effect when meerkats carrying Susu-DRB*13 were excluded from the model. As predicted, MHC diversity was not retained in the model (Fig. 5b and Supplementary Table 14). To underscore this, we assessed whether hosts with more alleles were also more likely to carry Susu-DRB*13. We found that a higher number of MHC alleles increased the chance of carrying Susu-DRB*13 ($G = 295.25$, chi-squared d.f. 6, $P < 0.001$; Fig. 5c) and homozygous individuals never carried haplotype F, whereas 10.7% of heterozygous individuals did. Although there are exceptions (for example, TB progression models), this pattern holds true across TB and fitness measures, as well as for supertypes and haplotypes.

Finally, we found that, when TB prevalence was high, implying a stronger selection by TB, the change in frequency of the allele Susu-DRB*13 from one year to the next tended to be positive (correlation: $R^2 = 0.51$, $P = 0.035$, Fig. 6), which implies individuals with this allele either survive better and/or produce more offspring. By contrast, Susu-DRB*03 was negatively correlated with TB prevalence ($R^2 = 0.52$, $P = 0.034$). Yet, even this time-lagged approach of selection-driven frequency changes cannot clearly distinguish between rare-allele advantage and fluctuating selection as the dominant mechanism driving evolution at the MHC without a comparison to other populations.

## Discussion

Tracking PMBS on host immunogenetics over time is a rare feat, with few studies able to show dynamics beyond one decade[28,29,44,45]. Particularly in the wild where other ecological and evolutionary processes co-occur, delineating between mechanisms driving PMBS proved difficult[6,9]. We collated data on two decades of selection by the TB-causing bacterium *Mycobacterium suricattae* in wild meerkats with molecular data from neutral and MHC genetic markers as well as socio-ecological and environmental information. We find that TB signs and TB-associated deaths drove stronger selection on MHC

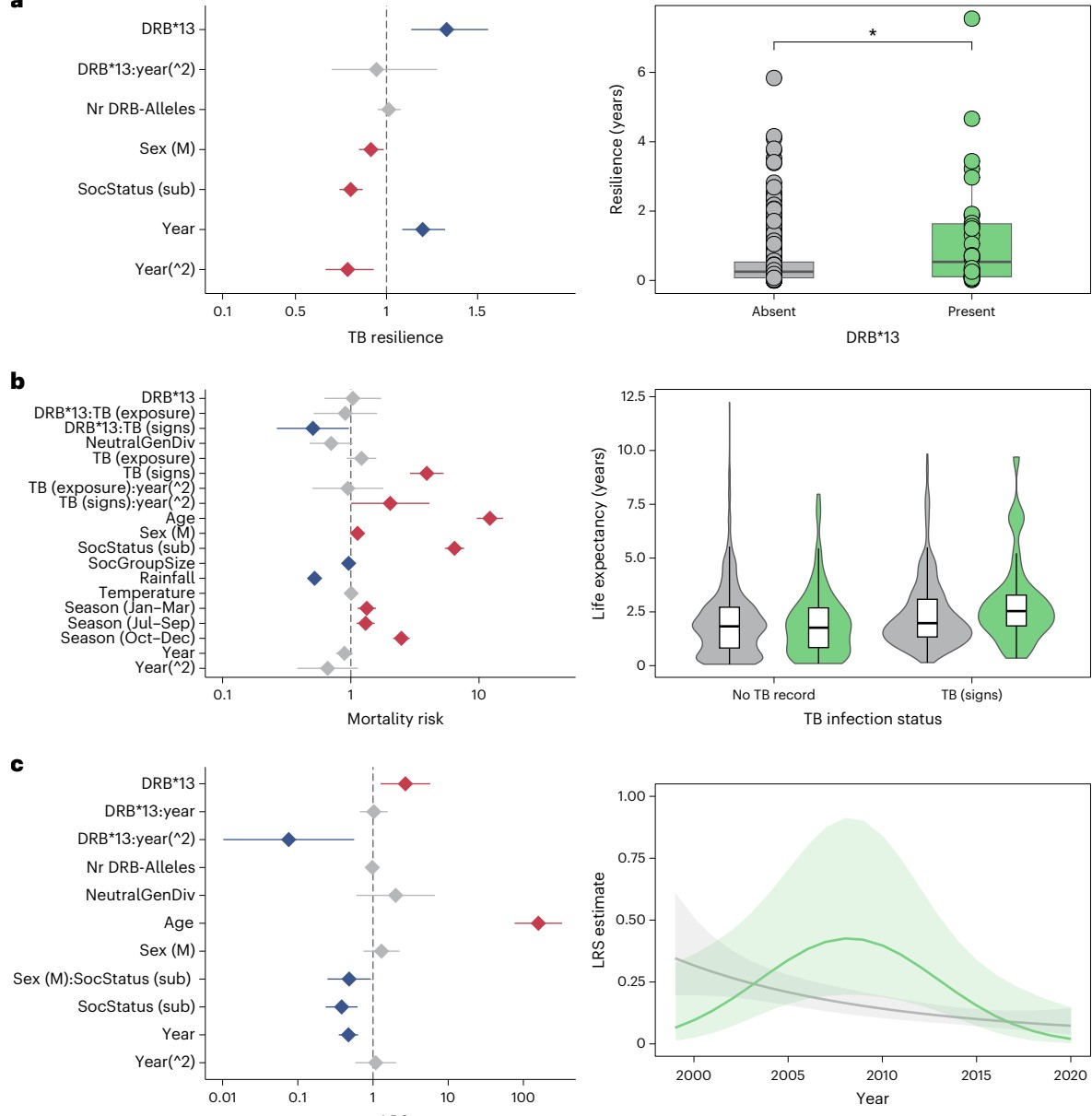

**Fig. 4 | MHC-DRB allele association with fitness. a–c**, Carrying MHC allele Susu-DRB*13 (green) when infected with TB improved TB resilience (*$P < 0.05$; $n = 1,497$) (**a**), life expectancy ($n = 1,535$) (**b**) and lifetime reproductive success (LRS; $n = 1,497$) (**c**). A summary of the generalized linear mixed-effects model output is given (left), and visualizations—including a box plot (median and interquartile range), a violin plot (density, median, and interquartile range) and the nonlinear effect of Susu-DRB*13 (green) over time—are presented (right). $P$-adjusted significant positive effects (±95% CIs; for example, longer survival with TB signs) are given in blue and negative effects (for example, shorter survival with TB signs) in red.

than on neutral genetic diversity. An initially susceptible MHC allele (and corresponding haplotype) emerged as advantageous variant in the later years, with benefits for host survival and reproduction. Our study offers rare evidence for multidecadal host immune gene evolution arising from selection by a devastating wildlife pathogen.

To understand pathogen-mediated selection on the MHC, one ideally compares MHC evolution with that of neutral genetic markers, examines MHC–pathogen associations in space and time across multiple populations, and aims to distinguish between the non-mutually exclusive mechanisms of PMBS, that is, rare-allele advantage, fluctuating selection and heterozygote advantage[9]. While our study meets some of these criteria, it falls short in others. For instance, the distribution of MHC allele frequencies was more even than for neutral markers, aligning with result from house finches[46] and montane

voles[47]. This pattern is indicative of PMBS and could arise from the advantage of having multiple co-dominant alleles, recycling of (rare) alleles in the population or fluctuating selection[9,19,27]. In addition, comparing population structure estimated from MHC and neutral genetic markers over time[48], we found a greater differentiation at the MHC. This was also found in one out of three hefts in the St. Kilda Soay sheep[29] and suggests either rare-allele advantage or fluctuating selection. Yet, our incomplete MHC and pedigree information for the 3,420 meerkats alive during our study, and our reliance on microsatellites over single-nucleotide polymorphisms, prevented a more sophisticated analytical approach to detect deviations in allele frequencies beyond neutral expectation (for example, gene-drop analysis)[45]. All in all, we find evidence that PMBS drives immune gene evolution in meerkats, albeit this conclusion is based on microsatellites.

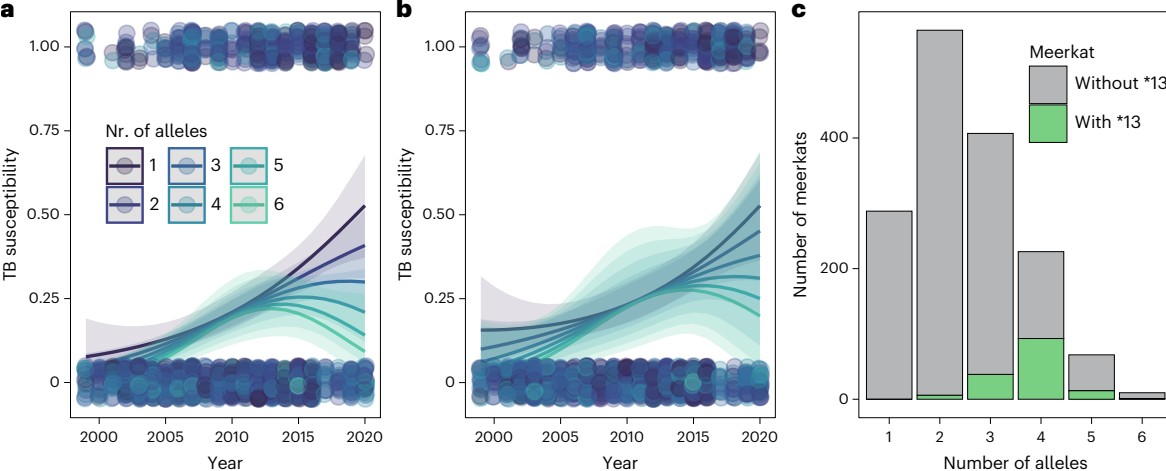

**Fig. 5 | Effect of MHC allelic diversity on TB susceptibility. a,b**, MHC allelic diversity impacts TB susceptibility (effect size ± 95% CIs) in the null model (not including MHC allele Susu-DRB*13 as an explanatory variable but including individuals with Susu-DRB*13 as part of their MHC profile) (**a**) but not after excluding individuals carrying Susu-DRB*13 (effect size ± 95% CIs) (**b**). **c**, The likelihood of carrying MHC allele Susu-DRB*13 increases in individuals with more MHC alleles.

We also found that clinical TB infection probability was associated with certain MHC alleles. For example, the risk of developing clinical signs was higher in meerkats carrying Susu-DRB*07 throughout the 20-year study. By contrast, meerkats with Susu-DRB*13 were initially more likely to develop signs than those with other MHC alleles, until around 2013, when the individuals with this allele became less likely to develop clinical TB. We found the same results for individuals carrying the haplotype F, to which Susu-DRB*13 was assigned to, although the analysis was carried out on a subsample. In humans as well, where TB has been a leading cause of death for centuries, individual single sequence variants between HLA-DQA1 and HLA-DRB1 have been strongly and repeatedly associated with *M. tuberculosis* infections and pulmonary TB disease in Icelandic, Russian and Croatian populations[34]. We lack this cross-populations comparison, which is necessary to differentiate between the mechanisms of PMBS[9]. This was attempted in an 11-year-long study in which three subpopulations of bank voles were sampled every 3–4 years[44]. The authors found that distinct MHC supertypes explained differences in helminth abundance between years and populations[44], invoking either locally (or regionally) fluctuating selection or rare-allele advantage as key mechanisms in this system. Yet, in line with predictions from the rare-allele advantage, voles carrying MHC supertypes common in the recent past were more intensely infected with helminths[44]. Applying this time-lagged approach, we showed that DRB*13 increased in frequency between years when TB prevalence was high in the year before, possibly suggesting a rare-allele advantage in our continuously sampled wildlife population.

A unique feature of our work is the switch from high to low susceptibility in the MHC allele Susu-DRB*13. When exposed to TB, meerkats with Susu-DRB*13 are also less likely to develop clinical TB. The fitness advantage was evident even among TB-positive meerkats, with those carrying Susu-DRB*13 surviving longer than individuals with other alleles in 90% of cases. Single MHC genotypes also determined survival in Seychelles warblers[49], red jungle fowls[50], Chiricahua leopard frogs[51] and root voles[52]. In rhesus macaques infected with simian immunodeficiency virus—which has a TB-like disease progression characterized by a long latency period followed by rapid deterioration once AIDS-like symptoms appear—a distinct MHC haplotype explained 48% of the observed variation in survival time[53]. Compared with this experimental study, the effects of MHC on meerkat survival in nature were subtle. This subtlety may reflect the complexity of innate and adaptive immune responses to TB: in humans, TB can evade or even exploit MHC-II-mediated responses, suppressing antigen presentation or co-opting CD4+ T cells to access host tissues[54–57]. As a result, strong immune activation may benefit the host only if the infection is rapidly cleared, which may be uncommon in natural settings[54,58]. Additionally, meerkats with Susu-DRB*13 benefitted from increased lifetime reproductive success. Associations between MHC genetics and reproductive output has been observed in stickleback[59], pied flycatchers[60], great tits[61] and Soay sheep[45]. As in these systems (but see ref. [61]), greater reproductive success probably arises from prolonged survival because living for longer is the strongest predictor of lifetime reproductive success. Greater resilience, prolonged survival and, as a consequence, increased reproduction may explain the steady increase in Susu-DRB*13 frequency when TB selection was high. Taken together, our findings support a role of host genetics in TB susceptibility and host fitness.

Neither the comparison with neutral markers, nor the patterns of MHC–pathogen association, nor the correlation between frequency changes and TB prevalence allowed us to fully discriminate between the mechanisms of PMBS. However, a few arguments support fluctuating selection and rare-allele advantage as key mechanisms at play in meerkats of the Kuruman River Reserve in South Africa. The greater divergence of MHC compared with neutral markers, the comparisons of models with and without Susu-DRB*13 and an increased chance to carry Susu-DRB*13 the more alleles an individual carried can be interpreted as exclusion criteria for the heterozygote advantage. Similar criteria ruled out heterozygote advantage in helminth-infected Soay sheep[29,45,62], in root voles infected with bacterial and protozoan blood parasites[52] and in humans infected with human immunodeficiency virus[63]. Still, we lack infection data for other pathogens affecting meerkats, which could reveal that different MHC alleles confer resistance to distinct pathogens, ultimately contributing to a heterozygote advantage[4,23,64]. Associating MHC measures and pathogen diversity would further help to delineate between PMBS mechanisms[9]. Moreover, we lack information on the co-evolution of *M. suricattae* over the course of our study. Because *M. suricattae*, like other members of the *Mycobacterium* complex, is likely to be under purifying selection, we expect a slow evolutionary rate[65]. This may allow ample time for hosts to adapt to a dominant variant and for beneficial genotypes to increase in frequency (as in some multigenerational studies[12,23,28,48] but not in others[16,17,62,66–69]). Although this suggests a rare-allele advantage, without a cross-population comparison, both rare-allele advantage and fluctuating selection are plausible.

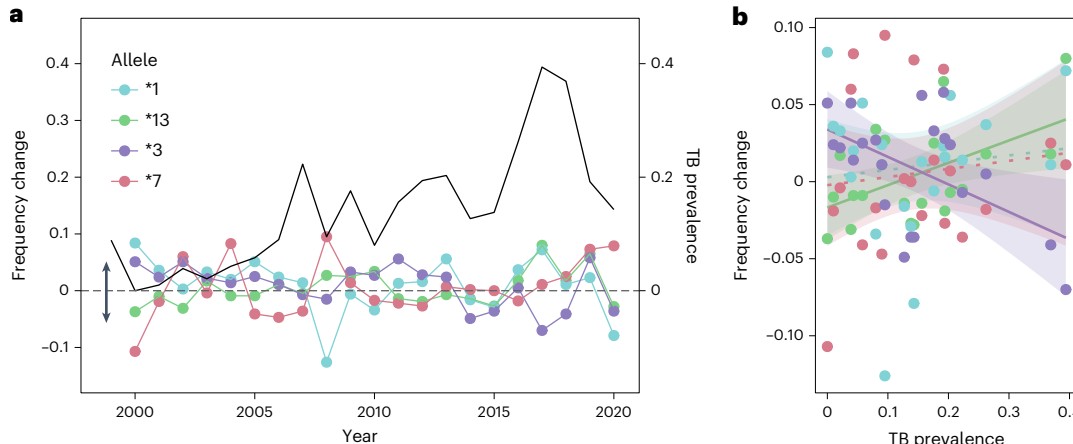

**Fig. 6 | MHC-DRB allele frequency changes with TB prevalence. a**, Visualization of the change in Susu-DRB allele frequency and TB prevalence (black line) over time. The dashed grey line represents zero change from one year to the next, with declines in the relative abundance of the MHC allele shown as negative changes (indicated by a downward-pointing arrow) and increases as positive changes (indicated by an upward-pointing arrow). **b**, Correlation (±95% CIs) between MHC-DRB allele frequency changes and TB prevalence. Dashed lines are non-significant, whereas solid lines indicate significant correlations.

However, socio-ecological factors cannot be ignored in a disease disseminated via direct contact. For instance, in the banded mongooses, which, like meerkats, deposit scent marking from their anal glands, the bacterial load of *M. mungi* in the anal gland and nasal planum was strongly linked to the level of tuberculous tissue in the lungs and lymph nodes[70]. Social communication may thus act as transmission route in meerkats too[41]. Equally important might be intragroup aggression, with the higher infection risk observed in the breeding pair potentially explained by their aggressive assertion of dominance in this cooperatively breeding species[37,41,71]. Intergroup aggression, female evictions or male dispersal may be other pathways by which TB spreads beyond one social group[41,72,73]. Yet, owing to the disease's long latency period, the individual circumstances leading to TB transmission are elusive even in this well-studied wildlife population. In parallel, higher temperatures and lower rainfall also increased the chance of TB outbreaks and mortality from TB in meerkats (our data and refs. 71,74; possibly mediated via a reduction in body condition[75] and a decline in gut symbionts[76]), mirroring observations from other study systems where drier environments are linked with increased TB susceptibility and prevalence[77,78]. Our data suggests that the allele Susu-DRB*13 was probably most beneficial when TB selection was strongest. For instance, shortly after a historic population collapse between 2012 and 2013 caused by a severe drought[79], TB signs had emerged in ~20% of the population[39]. In one scenario, the social groups that persisted may have been those containing more resilient and longer-lived individuals—possibly carriers of Susu-DRB*13—who were better able to support pup care and had more opportunities to reproduce[80,81]. Alternatively, in collapsing social groups, survivors may have disseminated both TB, especially as immigrants are more regularly harassed[82], and, thus, effectively spread TB during aggressive interactions. If they carried Susu-DRB*13, however, they may have profited from greater resilience, survival and, consequently, chances to reproduce and contribute their immunogenetic make-up to the new group. Future monitoring may answer whether Susu-DRB*13 has risen in frequency even further. In summary, our work illustrates the intricacies of contemporary evolution at the MHC driven by TB-driven selection in a longitudinally sampled wildlife population.

## Methods

### Study site and population
The meerkat population inhabiting the Kuruman River Reserve and surrounding farms (26° 580′ S, 21° 490′ E) in the Kalahari, South Africa, has been monitored since 1993[36]. The population includes several habituated social groups, where individual behaviour, life history events and health were recorded on a regular basis[83]. Tail tips were collected from all individuals born into the population at first emergence from the burrow[84,85]. We gathered first and final dates, social group membership, genealogy, lifetime reproductive success and records of clinical TB signs for 3,420 individuals born before 2020 from the Kalahari Meerkat Project Database (Supplementary Tables 1 and 2). We then focused on individuals observed at least 20 times, belonging to groups persisting for longer than 720 days with on average 6 or more individuals and a known TB history. Meerkats were classified as unexposed if no member of the group showed clinical signs of TB, as exposed if group members developed clinical signs of TB and as having clinical signs when lymph node swelling, lesions and physical deterioration were reported[39] (Fig. 1).

### Characterization of neutral genetic and MHC diversity
DNA was extracted from meerkat tail tips following the standardized Kalahari Meerkat Project protocol and used to genotype individuals at up to 18 microsatellite loci[84,85], with microsatellite data available for most individuals within the Kalahari Meerkat Project Database. Microsatellite data were used to estimate individual expected microsatellite heterozygosity ($H_{exp}$) with the R package 'GENHET'[86] and assign parentage with the R package 'relatedness'[84]. From parentage, we estimated lifetime reproductive success as the number offspring surviving to 1 year of age, at which juveniles reach sexual maturity.

To characterize the MHC class II, we developed and validated primers (CS1_Crcr_MHC2F: 3′-CCTGTSYCCACAGCACATTTCYT-5′; CS2_Crcr_MHC2R: 5′-GCTCAMCTCGCCGSTGCAC-3′) and successfully amplified MHC-DRB-exon 2 for 1,567 individuals following standard in-house procedures for high-throughput sequencing on an Illumina MiSeq platform (for details, see Supplementary Infomation). We used the ACACIA pipeline for quality filtering and allele calling[87], resulting in sequences of 247 bp. Three sequences identified included a 9-bp insertion, resulting in sequences of 256 bp. Our analysis included 360 technical replicates of 170 individuals, showing repeatability at individual level of 79.4% and repeatability at allele level of 96.1%.

### MHC class II gene annotation
We used reference-guided transcriptome assembly of 14 publicly available meerkat tissues (for details, see Supplementary Table 3) and using hisat2[88] aligned them to the current best meerkat reference genome (GCF_006229205)[89,90]. The MHC class II annotation was performed using blastn[91] with human class II query sequences against the

meerkat reference genome[89,90]. All BLAST hits were manually inspected along with aligned transcripts to determine correct splice sites and open reading frames. To determine which locus each allele belonged to, we compared identified DRB exon 2 alleles to reference sequence of each annotated DRB gene using blastn[92,93].

## MHC supertyping and allelic divergence

We clustered MHC alleles into supertypes based on shared physico-chemical properties at PSSs presumed to be involved in antigen binding[94–96]. PSSs were identified by comparing model 7 and model 8 in CodeML encoded in the program PAML4[97], which yielded 14 PSSs across the 37 alleles. After transcribing the amino acids at those 14 PSS into a matrix of $z$ values describing their physicochemical properties[98], we applied the find.cluster() and dapc() functions (that is, discriminant analysis of principal components) of the package 'adegenet'[99], and, based on goodness-of-fit assessment via $a$ scores and $x$ values, arrived at 12 clusters as most reliable supertype grouping. In addition to MHC supertypes, the divergence of MHC alleles (that is, mean and sum p-distance for amino acid alleles) was calculated using the DistCalc() function included in the 'MHCtools' package[100].

## MHC haplotyping

In contrast to supertyping, which is often performed to gain functional information and increase statistical power in MHC studies[101], haplotyping provides information on alleles inherited together, forming evolutionary units. MHC haplotypes were probabilistically inferred using the HapltFind() function in the 'MHCtools' package[100], designed to infer haplotypes by analysing the segregation of individual alleles from parents–offspring triads (POTs). Therefore, parentage was first assigned on the basis of the microsatellite data. This was done only for a total of 843 individuals with unambiguous parent–offspring information. These individuals grouped into 153 POTs consisting of both parents and their shared offspring across all litters. The output of the HpltFind() was then manually curated to exclude rare alleles (<8%) and two alleles (that is, Susu-DRB*07 and Susu-DRB*14) that were not inherited by offspring reliably and therefore could not reliably be assigned to haplotypes. Manual curating is necessary as the algorithm considers single POTs rather than including related POTs across multiple generations. If alleles occur in both parents, comparing inheritance patterns across different POTs and multiple generations can be necessary to disentangle haplotypes[100]. After reiterating Hapltfind(), we excluded POTs with unclear haplotype assignment, successfully inferring haplotypes for 741 meerkats.

## Statistics

Statistical analyses were performed using R (ref. [102]) versions 4.2.3. and 4.3.2 in RStudio[103], with plots generated using the ggplot2[104] and sjplot[105] packages. For all analyses, given the high number of control variables throughout, we chose an information criterion based approach, using the function dredge() of the 'MuMin' package[106] to identify the best models (delta AIC <2; code and data can be accessed publically[107]). We reported the full model average of all best models and report false discovery rate-corrected $P$ values to account for multiple testing[108] because we calculated models separately for each candidate MHC allele, supertype and haplotype associated with TB.

**Evolution at the MHC versus neutral markers.** Aside from calculating individual $H_{exp}$ as an explanatory variable in later analyses, we contrasted the evenness of MHC and microsatellite allele distributions and divergence over time to ascertain whether the MHC is indeed under selection. To contrast allele frequency distribution, we calculated the Simpson Evenness Index for both microsatellite and MHC data and compared the output using a paired $t$-test. A more even distribution (Simpson Index closer to 0) at the MHC than at neutral markers suggests overdominance and rare-allele advantage, whereas a lower

evenness (Simpson Index closer to 1) indicates directional and fluctuating selection[9]. To estimate molecular divergence, we estimated $\theta^\wedge$ of Fst calculated from each allele or locus for each pair of years and tested whether temporal and genetic distances were independent using a Mantel test[29]. The Fst values were then regressed against the number of years under the assumption that the temporal genetic divergence of MHC alleles should be higher than that found at neutral genetic markers, if selection varied over time[29]. The model included an interaction term between temporal distance and whether the value came from the MHC or microsatellite data. A significant interaction implies a difference in slope—and thus in selection—between the MHC and neutral markers.

**Co-occurrence analysis of MHC association with developing clinical signs of TB.** We used a probabilistic model of co-occurrence included in the 'cooccur' package[109] for each year to assess which MHC allele, supertype and haplotype was more or less frequently associated with TB than expected by chance. This and further analyses were computed on alleles, supertypes and haplotypes present in at least 5% of the study population (Supplementary Fig. 1b–d). Although filtering is necessary to maintain statistical power[101], this approach may lower chances to uncover links to recombinant or novel alleles arising within the study period. Furthermore, we restricted our analysis to years with more than ten TB-positive individuals (1999–2020), to MHC alleles and supertypes present in at least ten individuals, and to haplotypes present in at least five individuals. MHC alleles, supertypes and haplotypes with co-occurrence data for all years and multiple significant associations were considered as candidates for all follow-up analyses.

**Generalized linear model of MHC association with TB susceptibility, resilience and lifetime reproductive success.** To assess the impact of specific MHC alleles, supertypes and haplotypes and diversity on TB susceptibility (that is, ever developing clinical signs of TB), we ran a generalized linear model with binomial error structure, controlling for the biological factors sex, age and individual's neutral genetic diversity ($H_{exp}$)[60] as well as socio-ecological factors, namely dominance status (that is, whether an individual had ever attained dominance). MHC diversity metrics included the number of alleles and supertypes, mean and sum amino acid p-distance between alleles, and haplotype heterozygosity. To account for changes in TB prevalence throughout the study period and allow for a nonlinear effect of time in relation to MHC measures, we included individual death year as linear and quadratic term, and interactions between the year variables and MHC measures. Lifetime reproductive success was assessed considering a negative binomial error distribution while otherwise keeping the same model structure. The model assessing the factors influencing TB resilience (that is, number of years surviving with TB signs; log-transformed) excluded final age as explanatory variable and used a Gaussian distribution instead. To improve convergence, age, number of MHC alleles and supertypes, and year were rescaled. We excluded individuals with missing sex and microsatellite information and the only two individuals that carried seven MHC alleles. Analyses of alleles and supertypes were conducted on data from 1,497 individuals, while haplotype analyses used data from 714 individuals.

**Survival analyses of MHC association with TB progression and survival.** The temporal dynamics of TB progression and survival was modelled on 3-month periods (January to March, April to June, July to September and October to December) using the mixed-effects Cox model from the package 'coxme'[110]. For TB progression, models used the transition to clinical signs of TB before the start of the next period as endpoints. One model was run for each MHC allele, supertype and haplotype and MHC diversity measure, including only individuals observed at least five times during the respective period. We controlled

for biological and socio-ecological factors as described earlier and added population density (meerkats per squared kilometre), social group size, known environmental correlates of survival (that is, mean maximum temperature and mean rainfall; rescaled)[74,76] and season (January to March, April to June, July to September and October to December). Year was included as a linear and quadratic term, including the interactions between year and MHC measures. Meerkat ID was coded as a random factor. For modelling meerkat survival, we used the same model structure while including individual TB status (unexposed, exposed and signs) and the interactions between TB status and MHC measures with and without time to account for MHC effects on survival conditional on TB status and temporal shifts. After excluding individuals with missing metadata, the modelled datasets comprised 1,535 individuals and 12,936 meerkat periods for the TB progression analysis, and 13,804 meerkat periods for the survival analysis. From haplotyped individuals, we modelled with 722 individuals in 6,970 meerkat periods for the progression and 725 individuals in 7,492 meerkat periods for the survival analysis.

**Delineating between mechanisms of PMBS.** We are aware that irrefutable evidence for the distinct mechanisms of PMBS is achievable only in connection with a comparison across populations[9]. While we cannot provide this comparison, we have temporally dense, long-term data. A critical evaluation of MHC–pathogen associations over time, combined with comparisons to neutral evolution, can at minimum suggest the key PMBS mechanisms at play. First, we compared the fit of null models with models including specific MHC alleles, supertypes and haplotypes. This was done to test whether effects of MHC diversity (that is, number of MHC alleles and supertypes or haplotype heterozygosity) in null models are possibly driven by individual alleles. In cases where single alleles, supertypes and haplotypes appear linked to TB infection status, we additionally compared null models with and without individuals with the specific MHC allele, supertype and haplotype. Next, although a crude measure, we used a *G* test to determine whether specific alleles, supertypes and haplotypes were more likely to be found in meerkats with a higher number of alleles[52,63].

Finally, we calculated the change in MHC allele, supertype and haplotype frequency from one year to the next. Values below 0 imply the allele became less frequent, whereas values above 0 imply the allele become more frequent from one year to the next. We then correlated frequency change of candidate alleles with TB prevalence, as a measure of selection strength. The same time-lag approach was previously used in a short-lived vole population to account for selection in the previous year[44].

### Inclusion and ethics statement

Research for this study was conducted with permission of the ethical committee of 383 Pretoria University and the Northern Cape Conservation Service, South Africa (Permit 384 number: EC031-13, FAUNA 1020-2016). The field portion of our study was conducted at the Kalahari Meerkat Project, located at a remote site in the Kalahari Desert of South Africa. Fieldwork was discussed with and approved by the local stakeholders of the Kuruman River Reserve. As is the norm at the Kalahari Meerkat Project, fieldwork was conducted by an international and diverse team, including South Africa-born volunteers, to whom we are indebted.

### Reporting summary

Further information on research design is available in the Nature Portfolio Reporting Summary linked to this article.

## Data availability

All data used for statistical analyses are made publicly available via figshare at https://doi.org/10.6084/m9.figshare.26172985 (ref. 107). Source data are provided with this paper.

## Code availability

All code used for statistical analyses is publicly available via figshare at https://doi.org/10.6084/m9.figshare.26172985 (ref. 107).

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

## Acknowledgements

We thank A. Caranco and A. Akbal for the support with laboratory work and G. Camenisch for his logistic support during sample selection. We thank C. Duncan and M. Paniw for helpful discussions and sharing code, T. Vink for support with the database and of course all Kalahari Meerkat Project field managers and volunteers involved in meerkat data collection over the years. We thank V. Domingues for editorial support and two anonymous reviewers as well as J. N. Weber for their constructive feedback. The study was funded by German Research Foundation DFG SO 428/15-1 (S.S.); European Research Council 294494 (T.C.-B.); European Research Council 742808 (T.C.-B.); Human Frontier Science Program RGP0051/2017 (T.C.-B.); University of Zurich (M.M.); MAVA Foundation KRP 16026 (M.M. and T.C.-B.), the Swiss National Science Foundation and the Mammal Research Institute at the University of Pretoria, South Africa and Zoo Zürich. N.M.-K. received financial support for early career researchers from ProTrainU, Graduate and Professional Training Center at Ulm University.

## Author contributions

Funding acquisition: S.S., P.S.C.S., A.R., M.M. and T.C.-B. Study design and conceptualization: N.M.-K., D.W.M., S.S. and P.S.C.S. Data collection and curating: N.M.-K., M.M. and T.C.-B. Laboratory analyses: K.W., V.R., N.M.-K. and P.S.C.S. Statistical analyses: N.M.-K., D.W.M., V.R. and L.S. Paper writing: N.M.-K., D.W.M., A.R., P.S.C.S. and S.S. All authors commented and approved the submitted version of the paper.

## Funding

## Competing interests

The authors declare no competing interests.

## Additional information

**Correspondence and requests for materials** should be addressed to Nadine Müller-Klein.

[1]Institute for Evolutionary Ecology and Conservation Genomics, Ulm, Germany. [2]School of Science, Engineering, and the Environment, Salford University, Salford, UK. [3]Department of Evolutionary Biology and Environmental Studies, University of Zurich, Zurich, Switzerland. [4]Mammal Research Institute, University of Pretoria, Pretoria, South Africa. [5]Kalahari Research Centre, Kuruman River Reserve, Northern Cape, South Africa. [6]Large Animal Research Group, Department of Zoology, University of Cambridge, Cambridge, UK. [7]School of Life and Environmental Sciences, University of Sydney, Camperdown, New South Wales, Australia. [8]Australian Research Council Centre of Excellence for Innovations in Peptide and Protein Science, University of Sydney, Camperdown, New South Wales, Australia. [9]These authors contributed equally: Dominik W. Melville, Simone Sommer. ✉e-mail: Nadine.Mueller-Klein@uni-ulm.de

# Reporting Summary

## Statistics

For all statistical analyses, confirm that the following items are present in the figure legend, table legend, main text, or Methods section.

| n/a | Confirmed | |
|---|---|---|
| ☐ | ☒ | The exact sample size (*n*) for each experimental group/condition, given as a discrete number and unit of measurement |
| ☐ | ☒ | A statement on whether measurements were taken from distinct samples or whether the same sample was measured repeatedly |
| ☐ | ☒ | The statistical test(s) used AND whether they are one- or two-sided *Only common tests should be described solely by name; describe more complex techniques in the Methods section.* |
| ☐ | ☒ | A description of all covariates tested |
| ☐ | ☒ | A description of any assumptions or corrections, such as tests of normality and adjustment for multiple comparisons |
| ☐ | ☒ | A full description of the statistical parameters including central tendency (e.g. means) or other basic estimates (e.g. regression coefficient) AND variation (e.g. standard deviation) or associated estimates of uncertainty (e.g. confidence intervals) |
| ☐ | ☒ | For null hypothesis testing, the test statistic (e.g. *F*, *t*, *r*) with confidence intervals, effect sizes, degrees of freedom and *P* value noted *Give P values as exact values whenever suitable.* |
| ☒ | ☐ | For Bayesian analysis, information on the choice of priors and Markov chain Monte Carlo settings |
| ☒ | ☐ | For hierarchical and complex designs, identification of the appropriate level for tests and full reporting of outcomes |
| ☐ | ☒ | Estimates of effect sizes (e.g. Cohen's *d*, Pearson's *r*), indicating how they were calculated |

*Our web collection on statistics for biologists contains articles on many of the points above.*

## Software and code

Policy information about availability of computer code

| | |
|---|---|
| Data collection | Behavioural data on meerkats was collected using pendragon; Data was extracted from the Meerkat Database in June 2023; Database: MySQL |
| Data analysis | Bioinformatic and statistical analyses were performed in the software R (v4.2.3/ 4.3.2; R Core Team 2022). The R code including all packages used for the analyses of the current study will be made available on FigShare prior to publication, links to FigShare are provided with the submission. |

For manuscripts utilizing custom algorithms or software that are central to the research but not yet described in published literature, software must be made available to editors and reviewers. We strongly encourage code deposition in a community repository (e.g. GitHub). See the Nature Portfolio guidelines for submitting code & software for further information.

## Data

Policy information about availability of data

All manuscripts must include a data availability statement. This statement should provide the following information, where applicable:
- Accession codes, unique identifiers, or web links for publicly available datasets
- A description of any restrictions on data availability
- For clinical datasets or third party data, please ensure that the statement adheres to our policy

The dataset generated and/or analyzed during the current study will be made available on Figshare permanently pending acceptance. https://doi.org/10.6084/m9.figshare.26172985

# Research involving human participants, their data, or biological material

Policy information about studies with human participants or human data. See also policy information about sex, gender (identity/presentation), and sexual orientation and race, ethnicity and racism.

| Reporting on sex and gender | N/A |
|---|---|
| Reporting on race, ethnicity, or other socially relevant groupings | N/A |
| Population characteristics | N/A |
| Recruitment | N/A |
| Ethics oversight | N/A |

Note that full information on the approval of the study protocol must also be provided in the manuscript.

# Field-specific reporting

Please select the one below that is the best fit for your research. If you are not sure, read the appropriate sections before making your selection.

☐ Life sciences    ☐ Behavioural & social sciences    ☒ Ecological, evolutionary & environmental sciences

For a reference copy of the document with all sections, see nature.com/documents/nr-reporting-summary-flat.pdf

# Ecological, evolutionary & environmental sciences study design

All studies must disclose on these points even when the disclosure is negative.

| Study description | Data was collected within the Kalahari Meerkat Project, with data on known individuals collected since 1993. Life history data (births, deaths/ dissappearances, reproductive events), social group, group change events and clinical signs of Mycobacterium suricattae infections (TB) were recorded longitudinally on an individual level on an almost daily basis. Tissue samples for genetics were collected early in life from each individual born into the study group. Tissue samples were taken in from of a tail clip when meerkats were marked early in life. All other data used for this study was collected non-invasively. DNA had been extracted from tissue samples at the Universities of Edinburgh and Zurich. Microsatellite sequencing was completed at the University of Zürich. Aliquots were taken to Ulm University for MHC genotyping of the MHC-DRB locus via high throuput sequencing. Due to the longitudinal nature of the study, we could assess the effect of individual MHC-genotype on TB susceptibility (lifetime likelihood to develop signs of TB), TB resilience (time surviving after first signs of TB), progression (likelihood to progress to TB within 3 months) and overall mortality (likelihood to die within 3 months). Simultaneously, we could assess the link between frequency of specific MHC alleles and TB prevalence, testing for evidence for negative frequency-dependent selection as the mechanisms of pathogen mediated selection on wild meerkat MHC. |
|---|---|
| Research sample | Within the study population of wild, habituated meerkats, individuals from groups persisting for longer than 2 years with on average more than 6 resident individuals and a history of clinical infections with Mycobacterium suricattae were targeted for MHC-genotyping. Additionally, groups with no record of M. suricattae infections were targeted as controls. Our sampling regime resulted in 1567 successfully genotpyed individuals. Observational data and metadata for the study animals were extracted fom the Kalahari Meerkat Project Database. Tissue samples (tail tips) had been collected from individuals shortly after first emergence from the burrow and extracted DNA from the samples was utilized for the study. |
| Sampling strategy | Tissue samples were systematically collected, aiming at samples from all individuals within the study population. Within the Kalahari Meerkat Project, collection of observational data was distributed as evenly as possible across the study groups within the Project, with almost daily visits to each study group. |
| Data collection | TCB and MM organized the fieldwork, data management and sample collection, beginning in 1993. Data and samples were collected by a number of researchers and field assistants over the course of the project. KW, NMK and VR conducted lab work for MHC-genotpying, NMK, DWM and VR analysed the data. |
| Timing and spatial scale | Data collection within the Kalahari Meerkat Project has been ongoing from 1993 to date. For our study, we focussed on individuals born by 2020 and analysed data collected between 1999 and 2020. Data was collected from all habituated groups within the Kuruman Research Centre in Northern South Africa. |
| Data exclusions | Samples with less than 10.000 blasted reads were excluded from further analyses. Animals were excluded from analyses if metadata was incomplete (no information on sex, microsatellite heterozygosity). One individual carrying only one MHC-DRB allele with a premature stop Codon was excluded from further analyses. For linear modelling analyses, individuals that died before 1999 or after 2020 were excluded, reducing the dataset from 1566 individuals to 1497 for MHC-dataset and 714 individuals for the haplotyped dataset. |

| | |
|---|---|
| | For survial analyses, individuals with missing information and individual periods before 1999 and after 2020 were excluded, resulting in a final sample size of 1535 individuals. |
| Reproducibility | The extensive fieldwork that resulted in an enormous data set during the course of this study, allowing us to analyse almost half of the studied population of over 3400 individuals, spanning multiple social groups within an area of ~ 50 km^2 |
| Randomization | Due to longitudinal observational data collection, randomization is not applicable to the studied indiviuals. Individuals were assigned to social groups based on observed presence in the respective social groups. Infection status was determined based on clinical signs of infection for every individual on an almost daily basis. |
| Blinding | All laboratory work was performed blind to the identity of the sample. |

Did the study involve field work?  ☒ Yes  ☐ No

# Field work, collection and transport

| | |
|---|---|
| Field conditions | The Kuruman River Reserve covers around 50km2 of semi-arid dune country on either side of the dry bed of the Kuruman River in the South African Kalahari. The topography of the reserve consists of vegetated ("fossil") dunes covered with grasses and separated by valleys up to 300 m across. The dunes flatten as they approach the bed of the Kuruman River, which is usually dry. Vegetation consists of scattered camel thorn trees (Acacia erioloba) along the river bed, grading out into dry scrub dominated by annual and perennial grasses and Rhigozum scrub. There is around 250 mm/10 in of rain a year, mostly falling between October and March. In the summer (October-March), temperatures range from midday highs in the low 40s°C (November-December) to night-time lows of around 20°C. Temperatures are lower in winter (April-September) and often fall below 0°C at night in the coldest month (July). Humidity is typically very low. |
| Location | The Kalahari Research Centre is situated in the Kuruman River Reserve (KRR), in the Northern Cape, South Africa approximately 17 km south of the Botswana border and 30km west of the small town of Vanzylsrus. GPS coordinates: 26°58'S, 21° 49'E |
| Access & import/export | Samples utilized for the current study were already available from University of Zürich, where import/ export permits had previously been obtained. |
| Disturbance | Wild meerkats were carefully habituated and are familiar with the researchers, causing minimal disturbance to the population. |

# Reporting for specific materials, systems and methods

We require information from authors about some types of materials, experimental systems and methods used in many studies. Here, indicate whether each material, system or method listed is relevant to your study. If you are not sure if a list item applies to your research, read the appropriate section before selecting a response.

## Materials & experimental systems

| n/a | Involved in the study |
|---|---|
| ☒ | ☐ Antibodies |
| ☒ | ☐ Eukaryotic cell lines |
| ☒ | ☐ Palaeontology and archaeology |
| ☐ | ☒ Animals and other organisms |
| ☒ | ☐ Clinical data |
| ☒ | ☐ Dual use research of concern |
| ☒ | ☐ Plants |

## Methods

| n/a | Involved in the study |
|---|---|
| ☒ | ☐ ChIP-seq |
| ☒ | ☐ Flow cytometry |
| ☒ | ☐ MRI-based neuroimaging |

# Animals and other research organisms

Policy information about studies involving animals; ARRIVE guidelines recommended for reporting animal research, and Sex and Gender in Research

| | |
|---|---|
| Laboratory animals | The study did not involve laboratory animals. |
| Wild animals | Wild meerkats were observed on an almost daily basis, allowing for collection of detailed life-history data, including date of birth and death/ dissappearance. Individuals of all ages were included and data on their age, TB status and resident group were available. |
| Reporting on sex | Sex was included in all analyses as a control variable. Individuals were sexed based on external sexual characteristics (presence of testes). Of 1566 genotyped individuals, 701 were identified as female, 837 as male, and 28 did not have sex assignement (died as pups before sex determination) and were excluded from further analyses. |
| Field-collected samples | Tissue samples for genetic analyses were collected by cutting a small piece of the tail (~2 mm) from pups at first emergence from the burrow and from adult individuals when they immigrated into the study population. |

Sampled individuals returned to normal behaviour shortly after sampling and no infections of cuts were reported. There is no evidence for any short or long-term negative consequences of sampling the meerkats. Samples were stored in DMSO or 100 mm EDTA 95% ethanol at −20 °C. Extraction prior to 2002 was performed using standard chelex or phenol/chloroform methods. From 2002 on DNeasy Blood and Tissue kits by Quiagen were used for DNA extraction. Extracted samples were stored in double-distilled water at −20 °C, aliquots were transported to Ulm on ice.

**Ethics oversight**

Behavioural data and sample collection for this study was conducted with permission of the ethical committee of 383 Pretoria University and the Northern Cape Conservation Service, South Africa (Permit 384 number: EC031-13, FAUNA 1020-2016)

Note that full information on the approval of the study protocol must also be provided in the manuscript.

## Plants

**Seed stocks**

N/A

**Novel plant genotypes**

N/A

**Authentication**

N/A

