## [Peer Review File · Nature Ecology & Evolution]

Twenty-years of Tuberculosis-driven selection on meerkat MHC

Corresponding Author: Dr Nadine Mueller-Klein

Version 0:

Decision Letter:

12th August 2024

Dear Nadine,

Your Article, "Twenty-year co-evolutionary arms race between meerkat MHC and Tuberculosis" has now been seen by three reviewers. You will see from their comments copied below that while they find your work of considerable potential interest, they have raised quite substantial concerns that must be addressed. In light of these comments, we cannot accept the manuscript for publication, but would be very interested in considering a revised version that addresses these serious concerns.

We hope you will find the reviewers' comments useful as you decide how to proceed. If you wish to submit a substantially revised manuscript, please bear in mind that we will be reluctant to approach the reviewers again in the absence of major revisions.

If you choose to revise your manuscript taking into account all reviewer and editor comments, please highlight all changes in the manuscript text file in Microsoft Word format.

* Include a "Response to reviewers" document detailing, point-by-point, how you addressed each referee comment. If no action was taken to address a point, you must provide a compelling argument. This response will be sent back to the referees along with the revised manuscript.

* If you have not done so already we suggest that you begin to revise your manuscript so that it conforms to our Article format instructions at <http://www.nature.com/natecolevol/info/final-submission>. Refer also to any guidelines provided in this letter.

Link Redacted

If you wish to submit a suitably revised manuscript we would hope to receive it within 6 months. If you cannot send it within this time, please let us know. We will be happy to consider your revision so long as nothing similar has been accepted for publication at Nature Ecology & Evolution or published elsewhere.

Nature Ecology & Evolution is committed to improving transparency in authorship. As part of our efforts in this direction, we

are now requesting that all authors identified as 'corresponding author' on published papers create and link their Open Researcher and Contributor Identifier (ORCID) with their account on the Manuscript Tracking System (MTS), prior to acceptance. This applies to primary research papers only. ORCID helps the scientific community achieve unambiguous attribution of all scholarly contributions. You can create and link your ORCID from the home page of the MTS by clicking on 'Modify my Springer Nature account'. For more information please visit www.springernature.com/orcid.

Thank you for the opportunity to review your work.

[redacted]

Reviewer expertise:

Reviewer #1: host-pathogen co-evolution in natural populations of vertebrates, MHC

Reviewer #2: host-pathogen co-evolution in natural populations of vertebrates, MHC

Reviewer #3: host-pathogen co-evolution in natural populations of vertebrates

Reviewers' comments:

Reviewer #1 (Remarks to the Author):

In this paper, the authors studied the association between MHC variation, tuberculosis infection (TB), and survival in a long-term study of wild meerkats spanning two decades. It is one of the largest wild studies on MHC variation, and the excellent data enable the authors to observe the effects of MHC alleles/haplotypes on TB infection over a long period. They observed the cyclical change of MHC frequency. These results provide unique insights into the relevant field. However, I find some interpretations of the results somewhat confusing and over-interpreted. It is important to address the issues below properly before it is published in high-impact journals.

Major Issues:

1. Negative frequency-dependent selection (NFDS), fluctuating selection (FS), or heterozygote advantage (HA)?

The authors claim that the results present strong evidence of negative frequency-dependent selection. Unfortunately, I cannot agree. If they check their reference No. 8, Spurgin et al. suggested that "The best way to differentiate between rare-allele advantage and fluctuating selection would be to study MHC and neutral variation in relation to pathogen load over periods of evolutionary time in multiple replicate populations." I do not think the authors can theoretically differentiate NFDS from FS in a single population. That is why previous experimental studies demonstrating NFDS always used replicated populations (see refer No. 13 and No. 14 in your reference list). Similarly, in the Soay sheep paper (refer No. 40), they cannot clearly differentiate FS from NFDS. Indeed, NFDS does not necessarily act on rare alleles in terms of absolute frequency, and the advantage can occur with relatively rare frequency. However, the frequency of Susu-DRB*13 and haplotype F (not the rarest) further questions whether there is truly NFDS. It is clearly presented in Figure 2D that there is another haplotype experiencing a similar cyclical trend and the final allele frequency (AF) is more than 0.5.

In addition, the authors found the signature of HA regarding TB susceptibility but argued that this could be a by-product of NFDS, as individuals with more alleles are more likely to carry Susu-DRB*13. However, I think if they fit both the specific allele and the number of alleles in the same model and find both terms are significant, it suggests that both HA and NFDS/FS exist in the population.

Nevertheless, the results regarding the cyclical change of allele frequency are unique and solid. The authors need to change the text across the manuscript to reflect that FS and HA are also operating in this system.

2. Evidence of selection is not solid without lifetime breeding success

Natural selection is the differential survival and reproduction of individuals due to differences in phenotype. An increase in allele frequency as the selection response should only occur when the individuals carrying the favorable alleles have a high absolute fitness, defined as the total number of offspring throughout their lifetime. The authors have demonstrated the association between specific alleles/haplotypes and survival. However, lifetime breeding success has not been reported. Given that the pedigree is available in the current study, I think it is feasible and essential to examine the association between MHC alleles/haplotypes and lifetime breeding success (LBS). I understand there might be some individuals without a reliable record of LBS, which will impact the statistical power, but without LBS reported, it is not convincing to conclude that selection and selection response (change of allele frequency) are operating in this population.

3. Allele frequency change should be tested using pedigree data

In this paper, lines 116-119, the authors claimed that “TB prevalence among the MHC-typed individuals and MHC alleles, supertypes, and haplotypes fluctuated in frequency over time. In contrast, neutral genetic diversity, measured as expected microsatellite heterozygosity across up to 18 microsatellite loci, remained constant throughout the study period.” However, this was solely concluded from observation. I wonder if the authors can run simulations like Huang et al. (refer No. 40) along the pedigree to confirm the MHC allele frequency change is beyond the change under genetic drift. Also, they found a positive association between TB prevalence and Susu-DRB*13 frequency, but the results of other alleles and haplotypes are not reported. I think they need to demonstrate that the change of frequency in other alleles/haplotypes is not associated with TB prevalence. Finally, I would suggest the authors examine the correlation between allele/haplotype frequency and survival for a certain year. As TB infection is negatively associated with survival (fitness component), the correlation between survival and allele frequency will serve as good evidence to demonstrate that the cyclical change of frequency/haplotype is caused by selection.

4. Non-significant statistical results are not reported and no information regarding multiple testing

The authors used many statistical models, but there is no table summarizing the raw results. In this case, we do not know the exact p-value for each model. I assume the models presented in Figures 3-5 have also been used to test other alleles/haplotypes and probably found non-significant results. They need to report these results in the supplementary information. Also, although they ran models for each allele/haplotype, the multiple testing problem has not been addressed in this study. At least, Bonferroni correction should be used to examine whether the significant results are reliable under the multiple testing problem. I think it is important to present all their modeling results with sufficient details.

Minor Issues:

1. Unclear presentation of the plots

I found many plots in this manuscript quite confusing. For example, in Figures 2C and 2D, the color is too similar between alleles and haplotypes; it is nearly impossible to track the change of the alleles/haplotypes. I strongly suggest the authors use “linetype” or “linetype” plus “color” to differentiate them. For Figures 4B and 4D, we do not know what the color in the violin plot means exactly. I guess the grey one means individuals without the specific allele/haplotype, but it should be clearly explained in the figure legend.

2. Caveats of the study are not mentioned

I agree the data is amazing and there are very interesting results about the cyclical change of allele frequency and their potential link with NFDS or FS. However, there are still some caveats in this study. Even if those caveats may not necessarily bias the results, they should mention them in the paper. For example, the authors only used dozens of microsatellite loci to represent neutral diversity. Although they may not have the DNA and enough background genetic information of the meerkat to produce genome-wide genotyping, using solely microsatellite loci to represent neutral diversity is rather old-fashioned. Also, because of the limitation of amplicon-sequencing (relatively high sequencing error), they may have to exclude alleles with low frequency, but the study is therefore underpowered in detecting NFDS.

3. Lack of information about the study system

Meerkats are an interesting species with unique social structures and behaviors. However, I found the authors did not include enough information about their study system. I think they could include the following questions as table/figure/text in the supplementary for readers to better understand their study system.

What is the distribution of lifespan and lifetime breeding success for males and females, individuals included in the pedigree and without pedigree info (lifespan), MHC genotyped and not genotyped, TB infected and not infected?

How many groups of meerkats are observed each year, and what is the distribution of group size? What is the size of the study area?

If possible, also report the distribution of lifespan and LBS for dominant individuals and how they compare with other group members in terms of Q1.

Is migration between groups common in the study area?

Typo:

Supplementary information line 9, "Ffollowed" should be “followed”.

Reviewer #2 (Remarks to the Author):

The manuscript entitled “Twenty-year co-evolutionary arms race between meerkat MHC and Tuberculosis” presents an impressive study spanning 20 years of surveillance of wild colonies of meerkats, and tracking, among other aspects, the onset and course of Tb infections. Furthermore, the Authors present data interpreted as an evidence of NFDS acting on MHC genes in nature - a feat that had eluded researchers so far.

The manuscript is clear, well-structured and well-written, the presented figures are polished and appealing. While the overall picture drawn from the correlational data between tuberculosis susceptibility/resilience and changes in frequency of a

certain allele/ST/haplotype is suggestive and quite compelling, close look at the data presented brings out some questions and reveals certain weak points of the study. In particular, it rests heavily on correlations, which while indicative of a possible causative relationship - may not provide sufficient evidence to warrant publication in *Nature Ecology & Evolution*.

The story hangs mostly on the relationship between DRB*13 (and the associated supertype/haplotype) and TB. In particular, individuals carrying this allele initially faced higher TB infection probability, with the reversal of the effect over the course of the study. At that point, the allele seemed to increase in frequency. There are two major issues here:

1) For the story to hold we need to assume that, by chance, at the beginning, DRB*13 was a susceptible allele. The Authors argue that it was (l. 100-101) "increasing infection risk initially, but regaining resistance later after having decreased in frequency." Indeed, drop in frequency of DRB*13 following the time in which it was related to an increased TB susceptibility would support co-evolutionary arms race typical for NFDS dynamic. However, looking at the changes in frequency of DRB*13 (Fig. 3) – it seems to generally fluctuate, with little peaks every decade or so. Frequency at year 2000 was similar to that in 2010 (though see a later note on frequencies), followed by an increase by the end of the study (perhaps more visible for the haplotype than allele itself). This final increase in frequency, linked with resilience to TB, is presented as the main evidence of NFDS. However, there is no statistical evidence that this change was more than just a fluke or typical oscillation. In other words, was the change more pronounced than typically occurring fluctuations? Perhaps a comparison with the frequency trajectories of other alleles and/or microsatellite alleles which could serve as a neutral background could strengthen this claim. While the Authors say that the neutral genetic diversity, measured as expected microsatellite heterozygosity, was constant, particular microsatellite alleles likely change frequency over time.

2) Story of the DRB*13 fits the narrative nicely, but what about other alleles? What about alleles that always had high frequency and still seemed to be quite protective, e.g., DRB*1 – TB does not adapt to it? What about DRB*16 – it shows a reverse trend compared to DRB*13 – changes from being somewhat protective to susceptibility. This dynamic would be expected for an allele that used to be rare, rose in frequency and subsequently lost the selective advantage. Is it the case? Finally, what about the frequency of haplotype C given its adverse association with TB – it should drop in frequency? It is hard to gauge the changes in frequency from (a very pretty but not so easy to read) Fig 2D. Also, the sole allele of that haplotype is DRB*4 – but it was omitted from specific tests and graphic presentations. In the supplement (line 60), I found information that this allele was among the few that had "a premature stop codon at amino-acid position 82 and were thus presumed non-functional and excluded from subsequent analyses". I understand the logic behind this approach, but given what is seen and described for the haplotype C – a re-analysis would be warranted (perhaps for all non-functional MHC alleles) – as a non-functional MHC allele may indeed be a "bad news" during infection... At the same time, for a similar case of haplotype A containing only a non-functional DRB*2 - no such adverse effect is visible (based on Fig 2F).

Other, major points:

1) Calculation of frequencies of alleles (and haplotypes as well) – based on what is shown on Fig. 2C (and 2D) it seems that these are not really frequencies – as the values sum to more than 1. What is presented most likely represents "prevalence" of alleles, rather than frequency – i.e., proportion of individual that had given allele/haplotype? Given that the MHC genotyping methodology does not seem to allow assignment of alleles to loci (and so the Authors are dealing with a set of co-amplified variants per individual), looking at the prevalence and associating it with TB is one way to go – but it should be clearly explained and described on the figures. However, whether or not this measure can be "functionally" equated with frequency (as assumed in NFDS) – is another question entirely. For example, from the "adapt to evade" point of view, it can be a very different situation for a pathogen to have to deal with an MHC allele present in half of the population (prevalence = 0.5) but accompanied by several other alleles within each of the individuals, compared to a single locus situation. The notion that after several years of TB presence (and likely adaptation of this pathogen) higher individual MHC diversity started to translate to a decreasing TB susceptibility may hint to the significance of this distinction.

2) As signaled earlier - the study is purely correlational, and would be much stronger with some functional/experimental data to back it up. I am aware that adding more evidence (for example, data on a reciprocal TB genotype change, any experimental data on infection with the latest TB strain in animals with/without DRB*13, or at least some in silico modelling supporting superior binding of TB antigens to DRB*13) – is unlikely – but I feel it is a weakness that must be recognized, especially when aiming at a journal with such a broad impact as *Nature Ecology & Evolution*.

3) The long latency period of TB gives animals plenty of time to reproduce and pass on genes before eventual death – this fact was also noted by the Authors (l. 294-296). While they discuss this effect in relationship to dominant pairs, no information on whether TB affected (lifetime) reproductive success was provided – and this would be needed for a full picture of the TB effect on fitness. In general, a proof that TB enforces significant selective pressure to cause allele frequency changes over the course of the study is lacking. Even the number of generations or generation time was not provided for the species. Given availability of genealogies and detailed life histories, calculation of selection coefficients or similar metrics combined with appropriate simulations would strengthen presented findings.

4) The specific social & reproductive system of meerkats (which could be described in more detail), with social groups (of related/unrelated individuals?) and dominant pairs might also alter how the overall allele frequencies change in response to pathogen pressure (compared to what is expected under standard population genetics scenarios) – and this aspect should be discussed more directly.

5) Can the length of the latent phase in TB vary? If so, that might be a better feature to study the resilience of individuals

(akin to elite controllers in HIV/AIDS studies). I suppose that information was difficult to obtain – but might be more suitable for detection of protective MHC alleles. This aspect could be at least discussed in the article.

6) Measures of TB association with MHC alleles frequencies in a given year/year of death - given long latency, and presumably the retention of a TB strain throughout infection – would not the relationship with frequency of alleles at the time of infection be more informative? Again, I understand how this is unknown – but at present, the confounding factors include not only the time lag needed for parasites to adapt, but also the time lag between infection and actual development of observable symptoms – which further blurs relationship between alleles, their frequencies and susceptibility/resilience.

Minor comments:

- MHC expression – all genotyping is based on DNA analysis, are all (presumably functional = without premature stop codon) alleles expressed by the meerkats? While expression data for MHC class II is harder to get (as it requires preservation of tissues containing MHC II-expressing cells, like APCs, in an RNA-protective buffer), a smaller study showing overall pattern of expression would be useful. Given prominent birth-and-death dynamic within MHC locus, some alleles identified by amplification of exon fragments may really be non-functional, despite an ORF.

- Author argue that “our findings of a potential heterozygote advantage in later study years arose from the increased likelihood of carrying Susu-DRB*13 rather than MHC allelic and functional diversity per se.” – is the relationship with high number of alleles/STs maintained when carriers of DRB*13 allele are entirely excluded from an analysis? After all, both mechanisms can act in concert and are not mutually exclusive.

- Line 224: “in individuals” twice

Reviewer #3 (Remarks to the Author):

Summary:

By monitoring a total of 3,420 meerkats from the Kalahari, South Africa across a twenty year time span, the authors present an impressive dataset of tuberculosis infection dynamics (including onset of symptoms, progression, and eventually mortality), environmental covariates of disease (social and abiotic), and genetic variation. In particular, the paper examines both neutral genetic data from microsatellites and nucleotide variation in exon 2 of the MHC type 2 gene MHC-DRB. The goal of the study was to test for associations between MHC genetic diversity and disease, with the prediction being that coevolution would produce a signature of negative frequency dependent selection in host immune genes. The authors note that this mechanism of balancing selection has only rarely been observed in nature, likely due to the long span of time that is necessary to observe allele frequency and infection oscillations. Although the paper offers a number of interesting results, the main argument is that shifts in the frequency of a particular MHC allele (DRB*13), as well as the inferred haplotype containing this allele (haplotype F), are causally associated with shifts in tuberculosis best explained by negative frequency dependent selection.

Major comments:

The data, methods, and much of the main text are strong. This is indeed the longest term, largest (i.e. sample size), and most genetically detailed dataset that I've encountered related to the evolution of immunity in a natural vertebrate system. Many aspects of this study would be of interest to a large audience of biologists. I particularly commend the authors for their transparency in providing copious summaries of all the important data.

However, the authors primarily argue that negative frequency dependence drives the observed trends in MHC and disease variation. There are weaknesses to this interpretation that are not well addressed in the current manuscript. These limitations cast doubt on whether this is indeed a strong example of balancing selection, an argument that dominates the introduction and discussion sections of the manuscript. Below, I highlight my major concerns. When possible, I try to identify additional data, analyses, or arguments that might ameliorate these concerns. However, I'll emphasize from the start that the best way to address most of my critiques might be to simply reduce the manuscript's emphasis on balancing selection. The topic could still be addressed with less certainty, and this would open space for increased discussion of environmental correlates with disease, incorporation of some kind of pedigree or spatial information, and opportunities to reflect on both the strengths and weaknesses of the current dataset, including what kinds of data other researchers might want to collect before embarking on similarly ambitious projects.

Concerns:

1. The essence of the negative frequency dependent selection argument is that both host and pathogen populations are coevolving. Lines 75-78 argue that “robust evidence” only requires 3 types of data. The second type of data (b: strong pathogen selection) misses two additional requirements: showing that the pathogen population itself has evolved, and proving a causal connection between host and parasite alleles whose frequencies negatively covary over time. This study offers no data to support pathogen evolution or direct functional connections between host alleles infections resistance/susceptibility. Moreover, of the 4 important transitional states noted in Fig 1D, at best this paper only captures states 2 and 3. These limitations should be clearly stated. If there happen to be blood or tissue samples from the pathogen that could be used for genetic analyses, this would be an extremely important addition to the study that might be able to fill the pathogen side of this evidence gap.
2. In line with criticism 1, and with the author's own criteria (Lines 75-78), it's important to identify all possible reasons why host allele frequency oscillations might occur even in the absence of pathogen evolution. Although statistical analyses (Fig 2

E,F and Fig 3) do show significant associations between DRB*13 (or Haplotype F) and TB that change in direction over time, the trends are weak and could be biased by infection dynamics within particular social groups. For example, DRB*13 and HapF start at low frequencies and both drop to very low levels between 2002 and 2008. There really does need to be some discussion of how this rarity could influence false positive associations. For example, if those alleles reside in a social cluster that got hit hard by TB during those years then it would show an elevated signal of susceptibility driven more by rareness across social groups and stochasticity in infection prevalence than a functional connection to disease. I looked pretty carefully at the various statistical models and think the following would be important covariates to add:

-Social cluster ID

-Prevalence of infection (i.e., frequency of individuals with signs of infection, not just the presence/absence of infection in a group)

3. Connected to criticism 2, the discussion section highlights how the dominance system of meerkats might make them particularly susceptible to TB. I was intrigued by lines 294-299. If dominant individuals don't turnover in populations then it doesn't seem like there's much opportunity for the population to evolve new MHC alleles. This begs the question, how much evolution really could occur during the timespan of the study, and how exactly did the observed changes in DRB*13 take place? Given that the authors have data on group size over time, as well as some pedigree information on individuals (line 326 notes that relatedness was calculated for haplotype calling), it would be helpful to identify which social groups are driving the late increases in the DRB*13 allele, and describe what happened to those groups that allowed for evolution during the study. I'm particularly concerned that the trend of increased DRB*13 could come from a small number of groups with very low exposure rates late in the study.

4. In lines 144-155 the authors highlight TB associations that may be separate or jointly connected to effects of allele DRB*13 (or the related haplotype). The most relevant of these is the MHC diversity data: individuals with greater DRB diversity had greater TB resistance at the end of the study. The authors provide some data to suggest that this trend is driven by DRB*13 being present at high rates in MHC diverse individuals. These arguments are not persuasive:

a. Haplotype F happens to be one of a few that contains three distinct alleles. DRB*13 would thus necessarily be dragged along if selection was simply for diversity. This is not discussed in the paper.

b. Given the size of the datasets and the number of models tested, some effort should be made to compare the model and, ideally, identify which one best explains the infection data. There should at least be a description of R^2 values of models focusing only on DRB*13 or only on allelic diversity/heterozygosity. Another option might be to apply some kind of SEM/path analysis. There may not be enough breakdown in covariance between DRB*13 and allelic diversity to inform causal influences, but simply showing that there's a greater proportion of high allelic diversity individuals with DRB*13 than expected by chance is not compelling. If the authors choose to keep this argument, at very least the statistics need to be restricted to only those social groups where there was even a chance of inheriting the relatively rare DRB*13 allele.

c. Figure 6 builds on arguments that are not completely in line with negative frequency dependent selection, at least not across cycles of negative vs positive selection. It all depends on which stage in Fig 1D the process is occurring. During some points in time there should be a positive association between a causal allele and infection, and at others there should be no or negative correlations. The allele freq data for DRB*13 suggest that the study is observing stages 2-3 in Fig 1D. This distinction is important when making predictions about positive or negative correlations within vs between generations (Lines 205-216). The authors also don't provide much effort toward explaining the disconnect between looking at allele frequencies vs haplotype frequencies. Even with a smaller sample size there should still be some trace of an effect, unless the effect size is quite small.

5. Building on the finding of allelic diversity correlated with infection resistance, the topic of DRB functional supertypes gets rather short shrift in the main text. As I noted earlier, it's great that the authors include much of the ST data and models in the supp mat. But in Sup Fig 2 it appears that a number of other supertypes switch between positive and negative TB associations through the course of the study. The text only focuses on ST9 (containing DRB*13) and ST4. It is convenient that DRB*13 is in its own supertype, but given that supertype is probably a better predictor of functional similarity between alleles, there is an argument to be made for refocusing the paper on supertypes rather than individual alleles or haplotypes. If the authors choose to stick with haplotypes, additional justification should be provided.

7. Figures 4 and 5 are discussed only briefly in the results. The connections between TB resilience, life expectancy, disease progression and mortality are extremely interesting but not thoroughly synthesized in the discussion section. Why are each of these metrics important to add as main figures? Is it just in testing whether there's a genetic association with each of them? How do the genetic associations differ between metrics, even if only focusing on DRB*13, and why might this matter for understanding how evolution is affecting infection dynamics?

I greatly appreciate the effort that the authors made in conducting and writing up this interesting study. I hope that they find these comments to be relevant and useful.

-Jesse N. Weber

Version 1:

Decision Letter:

27th March 2025

Dear Nadine,

Your revised manuscript entitled "Twenty-year co-evolutionary arms race between meerkat MHC and Tuberculosis" has now been seen by the same three reviewers, whose comments are attached. The reviewers think the manuscript has improved in revision but they still have a number of important issues which will need to be addressed before we can offer publication in Nature Ecology & Evolution. We will therefore need to see your responses to the criticisms raised and to some editorial concerns, along with a revised manuscript, before we can reach a final decision regarding publication.

We therefore invite you to revise your manuscript taking into account all reviewer and editor comments. Please highlight all changes in the manuscript text file in Microsoft Word format.

* If you have not done so already please begin to revise your manuscript so that it conforms to our Article format instructions at <http://www.nature.com/natecolevol/info/final-submission>. Refer also to any guidelines provided in this letter.

* Extended Data Figures - please ensure that any supplementary figures and tables that are crucial to the manuscript's conclusions are converted into Extended Data figures and tables to increase visibility of these data. Extended Data figures and tables are online-only (present in the online PDF and full-text HTML versions of the paper), peer-reviewed display items that provide essential background to the article but are not included in the main article due to space constraints. A maximum of ten Extended Data display items (figures and tables) is permitted.

Link Redacted

Nature Ecology & Evolution is committed to improving transparency in authorship. As part of our efforts in this direction, we are now requesting that all authors identified as 'corresponding author' on published papers create and link their Open Researcher and Contributor Identifier (ORCID) with their account on the Manuscript Tracking System (MTS), prior to acceptance. ORCID helps the scientific community achieve unambiguous attribution of all scholarly contributions. You can create and link your ORCID from the home page of the MTS by clicking on 'Modify my Springer Nature account'. For more information please visit www.springernature.com/orcid.

[redacted]

Reviewers' comments:

Reviewer #1 (Remarks to the Author):

I appreciate the authors' efforts in addressing my concerns and suggestions, particularly in the discussion section. While I am satisfied with most of the revisions, I believe there are still some points that need to be addressed before the manuscript meets the standards of Nature Ecology & Evolution. Specifically, the authors should present all results comprehensively and include additional tables detailing sample sizes and phenotype distributions. Below, I outline my remaining concerns in relation to my previous comments:

1) Negative frequency-dependent selection (NFDS), fluctuating selection (FS), or heterozygote advantage (HA)?

The authors have made substantial rewording to better reflect this point. They have also provided a clear explanation of the

relationship between DRB*13 and MHC diversity. I am satisfied that they now acknowledge their inability to differentiate between different pathogen-mediated selection (PMS) mechanisms.

2) Evidence of selection is not solid without lifetime breeding success (LRS)

I appreciate the addition of models testing the association between MHC variation and LRS. However, I notice that the MHC-LRS association does not appear to be time-dependent or cyclical. This may be due to a lag between TB infection and LRS, which the authors could address in the discussion. Additionally, since they mention reproductive skew, they should provide a supplementary figure illustrating this. If the skew is significant, they could consider applying Bayesian modeling (e.g., MCMCglmm).

3) Allele frequency change should be tested using pedigree data

Unfortunately, the authors do not have sufficient pedigree data to perform a gene-drop analysis. However, they have plotted allele frequency changes against TB prevalence over time, which serves as a valuable supplement to their claims.

4) Non-significant statistical results are not reported, and there is no information regarding multiple testing corrections

This section still requires substantial improvement. In the supplementary tables, only some of the MHC alleles shown in Figure 2A (above the 5% threshold) are reported. For instance, results for DRB*06, DRB*10, and DRB*11 are missing. It is essential to provide a complete list of all eligible alleles, as well as corresponding results for supertypes and haplotypes.

5) Caveats of the study are not sufficiently discussed

In response to my previous comment, I believe the authors misunderstood my point about the study potentially being underpowered. While they have conducted thorough quality control to obtain reliable allele data, they have excluded low-frequency alleles (below 5%). Although this is a necessary step in amplicon sequencing to minimize sequencing errors, it also removes potentially novel alleles that could arise from recombination or mutations. Since low-frequency alleles are likely to be positively selected under NFDS, their exclusion may reduce the study's power to detect selection signals.

6) Lack of detailed information about the study system

I appreciate the addition of a sentence further describing the study system in the main text. Furthermore, Figure 2B and Supplementary Figure 1 address my previous concerns. However, I strongly recommend including a supplementary table to provide essential details about the sample size, as this is crucial for evaluating the study. The table should include:

Sample size for each year (total, MHC-genotyped, and TB-observed individuals)

Number of social groups per year

Distribution of key phenotypic traits, such as LRS for males and females separately

Reviewer #2 (Remarks to the Author):

I appreciate the effort the authors have made in thoroughly addressing the reviewers' comments. As the focus has shifted away from NFDS, I now find myself somewhat struggling to identify truly impactful findings. The authors propose a unique feature of their work was an observation of a switch from TB susceptibility to resistance in the MHC allele Susu-DRB*13. However, what remains unclear (even as a hypothesis) is the underlying mechanism driving this shift. This finding is intriguing, given that the allele was throughout the study of rather low abundance (including the time of susceptibility) and only began increasing in frequency once it started conferring resistance—a pattern that aligns with NFDS. Yet, if this allele was rare and associated with susceptibility, one could expect it to be lost from the population rather than to start to confer resistance? Or would the Authors argue that because it was not very common, pathogens “evolved away” and so it could become eventually protective? All is even more complicated, as the most abundant allele, *01 (which often accompanies the *13 on the F haplotype), appears to provide stable protection, as if pathogens did not adapt to it at all. While the findings are supported by an impressive dataset in a fascinating host-pathogen system, their broader implications remain somewhat limited. The most striking takeaway after all may be that DRB*13 was for a long time rare, but increased in frequency once it became protective. If the study had captured a full cycle of Red Queen dynamics (i.e., high frequency leading to susceptibility, a subsequent decline in frequency, followed by the acquisition of resistance and a resurgence in frequency), that would truly represent the Holy Grail of observational MHC studies. Also, what I find missing from the discussion—perhaps due to space constraints—is a reconciliation of some of somewhat conflicting observations surrounding the focal allele *13 (and its associated supertype or haplotype):

Line 154-155: “TB positive individuals carrying Susu-DRB*13 (and its corresponding ST09 and haplotype F) were more likely to progress from exposure to clinical signs of TB over the course of the study” – how this observation can be reconciled with supposedly protective effect the *13 allele conferred since 2010? They progressed but lived with it, suggesting some sort of control over infection>

Line 160-169: “Finally, individual lifetime reproductive success, as the ultimate Darwinian fitness measure, peaked between 2005 and 2013 in individuals that carried Susu-DRB*13”- it's interesting, as it seems to coincide with higher risk of TB infection faced by carriers of *13 until early 2010s?

Minor points:

Fig 2 – in caption description of panel E and D are mixed up.

Fig 3. Maybe I am misunderstanding or there is a mix-up – but I am not sure how there would be a non-linear relationship for Susu-DRB*13 if TB Hazard ratio on the lefthand side plot of B panel is significant for DRB*13:year but not DRB*13:year^[^2]? In contrast, the C panel shows significant effect of DRB*07:year^[^2], but the relationship is supposedly linear for DRB*07

(and so is shown on a righthand side of the C panel)?

Reviewer #3 (Remarks to the Author):

I thank the authors for their thoughtful and thorough response to my original critique. The largely rewritten manuscript provides important new data/insights (i.e., estimates of lifetime reproductive success, evidence that MHC alleles are expressed, neutral vs MHC genetic divergence, and more explanations of socio-environmental influences). The new organizing theme (disentangling the three causes of balancing selection) is also nuanced and more informative than the original focus. Despite the lack of a causal connection between MHC and disease, the paper still offers an impressive breadth of data and insightful analyses and discussion that would be of interest to NEE readers.

Although the paper has improved markedly, I still have a few concerns.

Major comments

1. The term “co-evolutionary arms race” in the title is not one of the main results from the paper and the title should be rewritten. While pathogen-host coevolution is important for maintaining MHC diversity, nothing in this article documents TB evolution. Any sentences suggesting evidence of coevolution (e.g., lines 38,214) should also be rephrased. That said, the authors do a much better job of noting throughout the manuscript that multiple selective forces can drive the apparent shifts in MHC allele frequency.
2. Comment #7 in my original review (biological story explaining patterns of selection) could still be more thoroughly addressed. I agree that lines 253-257 tell some of the story, and the final paragraph nicely restates the goal of excluding some types of selection. But I’m struggling to think of a straightforward explanation for how selection (either rare allele advantage or divergent selection) could explain observed changes with DRB*13. Here’s an example of how my logic keeps falling apart:
 - a. Given: DRB*13 correlates with increased susceptibility and resilience from 1999-2013. After that time, susceptibility decreases for the allele while resilience remains high. Also, reproductive fitness for individuals with the allele peak during the early period, but then drops to average levels for the whole population.
 - b. If the DRB*13 allele itself didn’t change in 2013 (i.e., no mutation/functional shift), then what could cause a decrease in susceptibility? This is where I’d really like to hear more biological predictions from the authors. I don’t think this shift in susceptibility could plausibly be explained by a rare-allele effect over such a short time span. A bit of evidence is given against heterozygote advantage and fluctuating selection, but there isn’t much critique of rare alleles driving this specific kind of pattern.
 - c. One could posit that TB has evolved. But there doesn’t seem to be a clear reason for the disease to evolve at this point. It is already increasing in population size by targeting/infecting the expanding number of DRB*13 individuals. Shouldn’t this increase TBs spreading potential given that those infected individuals are living longer.
 - d. Are there any general environmental/social changes that occur around the time of the susceptibility change? These are discussed as factors in the model, but I don’t remember them being tied to any particular years/time frames.
 - e. The only reasonable scenario I came up with is that DRB*13 containing individuals with clinical signs might be less likely to pass on the disease, leading to decreased prevalence (not necessarily susceptibility) in progeny of the same social group that contained the DRB*13 allele in later years.
 - f. The most important point here is that the paper mainly ends with generalities that may cause selection variation (lines 258-294). I applaud the authors for embracing the very difficult challenge of discriminating between selective forces. But, I don’t see how any of the listed explanations should drive the clear directional shift in DRB*13-TB susceptibility around 2013. I fully admit that this confusion may derive from my ignorance about the study system. A general reader would likely find this hypothetical context useful, and I couldn’t help but challenge the authors to think of a few more possible explanations.
3. I was confused by some of the word choice in the discussion. The authors may conflate the terms resistance and resilience at several places. I don’t think it’s primarily “resistance” (presence/absence of infection) that’s associated with the fitness benefit: lifetime reproductive success increases before 2013 while susceptibility is also increasing), but rather constantly highly resilience (lines 245-246). Granted, decreased susceptibility may contribute more in later generation. This distinction is important because some of the citations for MHC are specifically connected to resistance. Part of the problem here may be how I (and many others— See Bishop 2012 10.3389/fgene.2012.00168) think about pathogen resistance. This Bishop paper really does relate well to this work: could DRB*13 actively change selection on the pathogen (in which case it is resistance)? If so, at what time points? If not, then this is closer to tolerance. With the low levels of TB infection in meerkats, Bishop suggests that tolerance is difficult to understand theoretically, let alone measure.
4. While I like the new framework for the paper, Figure 1B may benefit from some editing. The allele freq distribution and population structure rows are nicely suited to this study, but the other two columns are a bit of a stretch. Pathogen load is presented as a quantitative trait, while here it is binary. And this study doesn’t address pathogen diversity. Perhaps this figure could also be used to highlight which of the patterns are tested in this paper, and which are supported?
5. The idea of connecting TB prevalence to change in relative prevalence of the DRB*13 allele is very good, but the statistics underlying Fig6B were unclear (text in lines 198-200, 434-436). The lagged approach that the authors are referencing asks if prevalence in the previous year predicts the amount of change in the next year. This is the most conservative approach, and the one I would favor. Why is the approach in this manuscript “similar to” (lines 434-436) and not just a “lagged approach”? Some of the methods text led me to think that the authors are testing whether prevalence in the current year is a predictor of change in allele prevalence from the previous to current year, or of changes within the current year.
6. Because this is a correlative study, please try to remove causal statements about the results (i.e., line 102 “determines” should be “correlates with”).

Minor comments:

1. Line 163: independent of "time", perhaps change to "year".
2. Line 250: I previously asked about the R^2 value for the models, and I appreciate the authors providing that info in their response. I don't necessarily think that the R^2 needs to be reported in this paper. But I can't help but revisit my previous effect size point. When citations like this one are made (single allele ~ 48% of variation SIV survival—a huge effect!), it begs the question of just how much of the variation in this study is explained by DRB*13. The model fits in the author's response letter (~11% and 17% for susceptibility and resilience, respectively) include both the DRB*13 allele and a few other factors, suggesting that the allelic correlations (or at least MHC correlations) are small relative to some other host-pathogen studies.
3. Lines 425-426: grammar
4. Line 420: "were" -> where
5. Line 452: Colored boxes represent alleles...
6. Fig 2 legend: captions D and E out of order.

As with the original submission, I hope the authors find these comments to be relevant and useful. This is important and well-done research. I appreciated the opportunity to read and learn from this manuscript.

-Jesse N. Weber

*****END*****

Version 2:

Decision Letter:

3rd June 2025

Dear Nadine,

Thank you for submitting your revised manuscript "Twenty-years of Tuberculosis-driven selection on meerkat MHC" (NATECOLEVOL-24071826B). It has now been seen again by two of the original reviewers and their comments are below. The reviewers find that the paper has improved in revision, and therefore we'll be happy in principle to publish it in Nature Ecology & Evolution, pending minor revisions to satisfy the reviewers' final requests and to comply with our editorial and formatting guidelines.

[redacted]

Reviewer #1 (Remarks to the Author):

Thank you very much for the revised version and I believe it addresses my concerns well. I am looking forward to seeing it published soon.

A minor suggestion: I recommend updating the text in Supplementary Tables 1 and 2 for clarity. Perhaps use "complete dataset" and "MHC-typed dataset" instead of "total" and "our subset" to avoid potential confusion.

Reviewer #3 (Remarks to the Author):

I thank the authors for carefully addressing the reviewers' comments. I have no further concerns about the content of this manuscript. It represents an impressive and important project that will be of great interest to readers of NEE.

We are pleased to submit a substantially altered version of the manuscript under the same title "Twenty-year co-evolutionary arms race between meerkat MHC and Tuberculosis" for a second round of peer-review.

We have addressed all comments raised in detail in our point-by-point responses below.

We want to briefly highlight key changes in response to the reviewers:

1. The reviewers asked for additional data and analyses where possible. This included information on meerkat life history, individual lifetime reproductive success and expression data. We are pleased to say that we added these pieces of information and this increased the data density of our work as well as the scientific rigor (e.g., lifetime reproductive success referenced in: L45, L168, L251, L397, Fig 4C, Supplementary Table 11, Supplementary Fig. 5). Dr. Luke Silver was added as a co-author over the course of this revision because he helped annotate the MHC region of the Meerkat genome, mapped our alleles to their respective loci, and validated their expression (referenced in: L109, L323 and following; Supplementary Material L74-121; Supplementary Tables 1, 3 and 4).
2. The reviewers asked to expand our comparison of neutral vs MHC diversity. This was now statistically tested using different approaches following suggestions in Spurgin and Richardson 2010 and applying established tests performed for long-term datasets (e.g., Charbonnel and Pemberton 2005; L263 and following). We found evidence for parasite-mediated balancing selection acting on the MHC and this resulted in a higher molecular divergence (L117, Fig 2E), while maintaining more even allele frequencies through time (L114, Fig 2D). Discussed in L215-229.
3. The reviewers requested to tone down the focus on NFDS selection and acknowledge methodological limitations that prevent us from conclusively differentiating between mechanisms of PMBS. We have shifted the focus of the manuscript to ask whether we can differentiate between mechanisms rather than arguing for NFDS (L100; L422 and following). Moreover, we added inferential evidence to delineate between mechanisms (L179 and following), while acknowledging our shortcomings throughout (L202, L79) and most clearly in the discussion (L238, L273, L291).

Importantly, even after altering some analyses based on reviewer suggestions and when adding new supporting data (e.g., lifetime reproductive success), the main conclusions remain. Reviewer 1 puts it as “[The] results regarding the cyclical change of allele frequency are unique and solid,” and we agree that this remains a unique finding and provides novel insights in the role of host genetics in wildlife species surveyed for well over 20 years.

Kind regards and best wishes,

Nadine Müller-Klein (*on behalf of all authors*)

Detailed responses to Reviewer comments:

Reviewer expertise:

Reviewer #1: host-pathogen co-evolution in natural populations of vertebrates, MHC

Reviewer #2: host-pathogen co-evolution in natural populations of vertebrates, MHC

Reviewer #3: host-pathogen co-evolution in natural populations of vertebrates

Reviewers' comments:

We thank all reviewer for their thorough peer review and the constructive comments on our manuscript. We are glad the manuscript was received overwhelmingly well. We have carefully read through the comments and incorporated suggestions, including adding lifetime reproductive data, mapping the identified alleles to the MHC region of the meerkat genome, and comparing the effect of neutral vs MHC evolution over time. We have also toned down some of our claims or provided additional evidence for others. These changes have certainly improved the clarity and scientific accuracy of our manuscript. In cases where requests were not feasible with our data we gave a detailed reasoning, and acknowledge the limitations in the discussion. Because the changes were so substantial we did not highlight changes in the MS, but rather refer to line numbers for specific changes, and hope the reviewers enjoy reading this manuscript that they helped to improve.

Please find all detailed responses below.

Reviewer #1 (Remarks to the Author):

In this paper, the authors studied the association between MHC variation, tuberculosis infection (TB), and survival in a long-term study of wild meerkats spanning two decades. It is one of the largest wild studies on MHC variation, and the excellent data enable the authors to observe the effects of MHC alleles/haplotypes on TB infection over a long period. They observed the cyclical change of MHC frequency. These results provide unique insights into the relevant field. However, I find some interpretations of the results somewhat confusing

and over-interpreted. It is important to address the issues below properly before it is published in high-impact journals.

Major Issues:

1. Negative frequency-dependent selection (NFDS), fluctuating selection (FS), or heterozygote advantage (HA)?

The authors claim that the results present strong evidence of negative frequency-dependent selection. Unfortunately, I cannot agree. If they check their reference No. 8, Spurgin et al. suggested that “The best way to differentiate between rare-allele advantage and fluctuating selection would be to study MHC and neutral variation in relation to pathogen load over periods of evolutionary time in multiple replicate populations.” I do not think the authors can theoretically differentiate NFDS from FS in a single population. That is why previous experimental studies demonstrating NFDS always used replicated populations (see refer No. 13 and No. 14 in your reference list). Similarly, in the Soay sheep paper (refer No. 40), they cannot clearly differentiate FS from NFDS. Indeed, NFDS does not necessarily act on rare alleles in terms of absolute frequency, and the advantage can occur with relatively rare frequency. However, the frequency of Susu-DRB*13 and haplotype F (not the rarest) further questions whether there is truly NFDS. It is clearly presented in Figure 2D that there is another haplotype experiencing a similar cyclical trend and the final allele frequency (AF) is more than 0.5.

Authors response: We thank the reviewer for their feedback. We agree that we cannot differentiate between rare-allele advantage and frequency dependent selection using this one study population (as clearly outlined in Spurgin & Richardson). We openly discuss this caveat in the discussion now (L238, L273, L291), but also throughout the manuscript (L202, L79).

Nevertheless, we think our manuscript now offers multiple lines of evidence that selection is acting on Susu-DRB*13 (L179 and following; see responses to other reviewers later).

We agree with the reviewer that even relatively rare alleles might become advantageous. We now also go more into the results for the Allele Susu-DRB*07 (L148; Fig 3C) and Allele

Susu-DRB*1 (L170). The latter is more common, and shows the opposite pattern to Susu-DRB*13. Similarly, DRB*3 showed changes in allele frequency with TB selection pressure (Fig 6B, even though no clear connection to TB resistance or host fitness was detected).

We also added other lines of evidence that the effect may be driven by MHC-DRB*13, by comparing models with and without DRB13 (see later responses). However, since we cannot rule out other mechanisms, we changed the phrasing throughout the manuscript to indicate that we cannot differentiate between FS and NFDS (e.g., L291).

In addition, the authors found the signature of HA regarding TB susceptibility but argued that this could be a by-product of NFDS, as individuals with more alleles are more likely to carry Susu-DRB*13. However, I think if they fit both the specific allele and the number of alleles in the same model and find both terms are significant, it suggests that both HA and NFDS/FS exist in the population.

Authors response: We agree with the reviewer that we cannot entirely rule out the contribution of HA in our study population (particularly since we have an incomplete list of pathogens affecting this population, and “heterozygote advantage may be best understood when considered in the context of multiple pathogens, as MHC alleles conferring resistance to one pathogen can increase susceptibility to another” Spurgin & Richardson 2010, which we acknowledge in L281). The reviewer is also right that if the number of alleles and the specific allele were included in the model and showed significant associations with TB both HA and NFDS/FS likely act on the population and are important in TB resistance. However, this is not the case. MHC diversity was never significantly associated with TB susceptibility, resilience and survival if allele DRB*13 was included in the model (Supplementary Table 8, 10, 11). Number of alleles was only significant when Susu-DRB*13 was not included in the model.

We recognize that this is not enough to disentangle the frequency-dependent component from heterozygote advantages in TB mediated selection found in our population. In the original and current MS, we addressed this using a G-test in order to determine whether the likelihood of having Susu-DRB*13 increased with number of alleles or whether heterozygous

individuals were more likely to have haplotype F (according to Kloch et al. 2013 and Arora et al. 2020). This is the case (Fig 5C; L194), but we understand that this is not enough. In addition, we re-ran models for TB susceptibility and resilience in a subset of individuals not carrying allele Susu-DRB*13 (L189 and following; Supplementary Table 13). The non-significant results indicate that, at least for TB-mediated selection, the effect of HA is limited.

We phrase these findings now as inferential evidence rather than ruling HA out (L180 and following), and this is discussed at length in the discussion (L273-283)

Nevertheless, the results regarding the cyclical change of allele frequency are unique and solid. The authors need to change the text across the manuscript to reflect that FS and HA are also operating in this system.

Authors response: Thank you for underscoring that the cyclical change in allele frequency is unique. We rewrote parts of the introduction, results and discussion (e.g., L102, L202, L273-294) to emphasize that FS and HA are possible contributors that we cannot fully rule out.

2. Evidence of selection is not solid without lifetime breeding success

Natural selection is the differential survival and reproduction of individuals due to differences in phenotype. An increase in allele frequency as the selection response should only occur when the individuals carrying the favorable alleles have a high absolute fitness, defined as the total number of offspring throughout their lifetime. The authors have demonstrated the association between specific alleles/haplotypes and survival. However, lifetime breeding success has not been reported. Given that the pedigree is available in the current study, I think it is feasible and essential to examine the association between MHC alleles/haplotypes and lifetime breeding success (LBS). I understand there might be some individuals without a reliable record of LBS, which will impact the statistical power, but without LBS reported, it is not convincing to conclude that selection and selection response (change of allele frequency) are operating in this population.

Authors response: Thank you very much for bringing this up. We indeed have this information. In the process of writing this we had opted to retain this information for a

follow-up study, but we are more than happy to follow the reviewers request and add LRS to the MS. Specifically, LRS was calculated as number of offspring surviving to 1 year of age (at which point the juveniles reach sexual maturity; L313). We then modelled links between MHC diversity and composition and lifetime reproductive success using the same model structure as when modelling associations with TB (L397). We find that, even in spite of the strong reproductive skew towards dominant individuals and a strong effect of longevity on reproductive success (Supplementary Table 12), both DRB*01 and DRB*13 affect LRS. We report our findings in the results (L168-173) and discuss subsequently (L251-256).

3. Allele frequency change should be tested using pedigree data

In this paper, lines 116-119, the authors claimed that “TB prevalence among the MHC-typed individuals and MHC alleles, supertypes, and haplotypes fluctuated in frequency over time. In contrast, neutral genetic diversity, measured as expected microsatellite heterozygosity across up to 18 microsatellite loci, remained constant throughout the study period.”

However, this was solely concluded from observation. I wonder if the authors can run simulations like Huang et al.(refer No. 40) along the pedigree to confirm the MHC allele frequency change is beyond the change under genetic drift. Also, they found a positive association between TB prevalence and Susu-DRB*13 frequency, but the results of other alleles and haplotypes are not reported. I think they need to demonstrate that the change of frequency in other alleles/haplotypes is not associated with TB prevalence. Finally, I would suggest the authors examine the correlation between allele/haplotype frequency and survival for a certain year. As TB infection is negatively associated with survival (fitness component), the correlation between survival and allele frequency will serve as good evidence to demonstrate that the cyclical change of frequency/haplotype is caused by selection.

Authors response: Thank you for these suggestions. We had tried to implement the genedrop analysis performed by Huang et al., and actually been in contact with the developer Susan Johnston about to make it work for our data. While the pedigree is quite dense compared with the Soay sheep population the pedigree remains incomplete, and the gaps in MHC-information renders the data too sparse to run the genedropper function on the smaller subset individuals typed for their MHC. However, there are alternative ways to

compare neutral vs MHC evolution and we applied them in our updated manuscript. For one and this is in line with your suggestion looking at frequency change over time, we followed the pipeline of Charbonnel and Pemberton (2005), who compared neutral vs MHC divergence over time in the Soay sheep population. Using this approach, we can showcase statistically that the divergence (or differentiation) over time was steeper at the MHC than at the microsatellite markers (Figure 2E). Additionally, we show that MHC allele frequency distributions are more even than those of microsatellite alleles (Fig 2D).

Additionally, neutral genetic diversity, represented by an heterozygosity index, was always included in all our models, following the rationale by (Canal *et al.* 2024), and reported.

We also show the requested relationship between frequency change and TB prevalence over time for the candidate alleles (Fig 6).

Lastly, the link to survival was already presented in the MS and can now be found in L163-168; Fig 4B; Supplementary Table 11) and is discussed in L246-251 and also in L268-272.

4. Non-significant statistical results are not reported and no information regarding multiple testing

The authors used many statistical models, but there is no table summarizing the raw results. In this case, we do not know the exact p-value for each model. I assume the models presented in Figures 3-5 have also been used to test other alleles/haplotypes and probably found non-significant results. They need to report these results in the supplementary information. Also, although they ran models for each allele/haplotype, the multiple testing problem has not been addressed in this study. At least, Bonferroni correction should be used to examine whether the significant results are reliable under the multiple testing problem. I think it is important to present all their modeling results with sufficient details.

Authors response: All raw models were presented in the supplementary information and other reviewers had taken reference to them. We have updated the section to also include models run on LRS for this submission (Supplementary Table 8-12). From these raw model results, the reviewer can also see that all models were false-discovery rate adjusted, and we

state this in the statistics section and each table header (e.g., L135, L360 and found in the Table headers). We apologize if this was not clear in the first version.

Minor Issues:

1. Unclear presentation of the plots

I found many plots in this manuscript quite confusing. For example, in Figures 2C and 2D, the color is too similar between alleles and haplotypes; it is nearly impossible to track the change of the alleles/haplotypes. I strongly suggest the authors use “linetype” or “linetype” plus “color” to differentiate them. For Figures 4B and 4D, we do not know what the color in the violin plot means exactly. I guess the grey one means individuals without the specific allele/haplotype, but it should be clearly explained in the figure legend.

Authors response: Thank you for your feedback. We updated all plots to improve clarity.

2. Caveats of the study are not mentioned

I agree the data is amazing and there are very interesting results about the cyclical change of allele frequency and their potential link with NFDS or FS. However, there are still some caveats in this study. Even if those caveats may not necessarily bias the results, they should mention them in the paper. For example, the authors only used dozens of microsatellite loci to represent neutral diversity. Although they may not have the DNA and enough background genetic information of the meerkat to produce genome-wide genotyping, using solely microsatellite loci to represent neutral diversity is rather old-fashioned. Also, because of the limitation of amplicon-sequencing (relatively high sequencing error), they may have to exclude alleles with low frequency, but the study is therefore underpowered in detecting NFDS.

Authors response: We thank the reviewer for their assessment. We agree that microsatellites are “old-fashioned”. Their inclusion is founded in history (Griffin et al. 2005) and decades of research based on this data (e.g., Spong et al. 2008). While there are efforts underway to type meerkats using SNPs an insufficiently large proportion of the population

has been covered so far. We are transparent about this shortcoming in the new version (L226-229).

We disagree with the reviewer on the notion that our study is underpowered. We undergo very stringent allele filtering steps (outlined in the supplementary material and methods) to avoid erroneously allele calling sequences arising from sequencing errors. Additionally, we state clearly in the methods (L381-386) and results (L126) that rare alleles/haplotypes/supertypes (in <5% of the population) were excluded from any further analysis. We only analysed years with a sufficient number of MHC typed individuals and TB cases and only considered alleles that met those thresholds in all years. We consider this quite a conservative approach.

What we do not have is data from multiple population, which is a shortcoming and limits our ability to delineate NFDS from other possible mechanisms. We debate this in detail in the updated discussion (e.g., L273).

3. Lack of information about the study system

Meerkats are an interesting species with unique social structures and behaviors. However, I found the authors did not include enough information about their study system. I think they could include the following questions as table/figure/text in the supplementary for readers to better understand their study system.

What is the distribution of lifespan and lifetime breeding success for males and females, individuals included in the pedigree and without pedigree info (lifespan), MHC genotyped and not genotyped, TB infected and not infected?

How many groups of meerkats are observed each year, and what is the distribution of group size? What is the size of the study area?

If possible, also report the distribution of lifespan and LBS for dominant individuals and how they compare with other group members in terms of Q1.

Is migration between groups common in the study area?

Authors response: We thank the reviewer for this very good suggestion. We are delighted to provide the requested information on meerkat ecology and life history (L85-95). Specifically,

we go into more detail on their ecology in relation to the spread of TB (L87 and following) and dedicate an entire paragraph in the discussion to the topic (L258-272). Moreover, we chose to visually present data the requested life history traits, which can be found in Supplementary Figure 1.

Typo:

Supplementary information line 9, "Ffollowed" should be "followed".

Authors response: Thanks. This was corrected.

Reviewer #2 (Remarks to the Author):

The manuscript entitled "Twenty-year co-evolutionary arms race between meerkat MHC and Tuberculosis" presents an impressive study spanning 20 years of surveillance of wild colonies of meerkats, and tracking, among other aspects, the onset and course of Tb infections. Furthermore, the Authors present data interpreted as an evidence of NFDS acting on MHC genes in nature - a feat that had eluded researchers so far.

The manuscript is clear, well-structured and well-written, the presented figures are polished and appealing. While the overall picture drawn from the correlational data between tuberculosis susceptibility/resilience and changes in frequency of a certain allele/ST/haplotype is suggestive and quite compelling, close look at the data presented brings out some questions and reveals certain weak points of the study. In particular, it rests heavily on correlations, which while indicative of a possible causative relationship - may not provide sufficient evidence to warrant publication in Nature Ecology & Evolution.

The story hangs mostly on the relationship between DRB*13 (and the associated supertype/haplotype) and TB. In particular, individuals carrying this allele initially faced higher TB infection probability, with the reversal of the effect over the course of the study. At that point, the allele seemed to increase in frequency. There are two major issues here:

1) For the story to hold we need to assume that, by chance, at the beginning, DRB*13 was a

susceptible allele. The Authors argue that it was (l. 100-101) “increasing infection risk initially, but regaining resistance later after having decreased in frequency.” Indeed, drop in frequency of DRB*13 following the time in which it was related to an increased TB susceptibility would support co-evolutionary arms race typical for NFDS dynamic. However, looking at the changes in frequency of DRB*13 (Fig. 3) – it seems to generally fluctuate, with little peaks every decade or so. Frequency at year 2000 was similar to that in 2010 (though see a later note on frequencies), followed by an increase by the end of the study (perhaps more visible for the haplotype than allele itself). This final increase in frequency, linked with resilience to TB, is presented as the main evidence of NFDS. However, there is no statistical evidence that this change was more than just a fluke or typical oscillation. In other words, was the change more pronounced than typically occurring fluctuations? Perhaps a comparison with the frequency trajectories of other alleles and/or microsatellite alleles which could serve as a neutral background could strengthen this claim. While the Authors say that the neutral genetic diversity, measured as expected microsatellite heterozygosity, was constant, particular microsatellite alleles likely change frequency over time.

Authors response: We thank reviewer 2 for their careful assessment of our work, the praises of our writing and sequencing effort. While we agree that our study is correlational in nature (and we acknowledge this; L274), we are convinced that the updated version further consolidates why our work will stand out among MHC literature and as ecological and evolutionary long-term study.

The reviewer is correct that evidence for NFDS is largely dependent on the findings concerning Susu-DRB*13 (although there are several other reported MHC effects; e.g., L148). We disagree with the reviewers assessment, however, that Susu-DRB*13 fluctuated heavily. In fact, during the TB susceptibility period in the early 2000s the alleles frequency is low and consistently so (Fig 2C). The allele undergoes an increase in frequency when it becomes increasingly associated with resistance (and the reviewer correctly states that this is even clearer for the haplotype) and when pathogen pressure is high (Fig 6). Our model also underscores that this change is significantly different compared to meerkats without Susu-DRB*13 (e.g., Supplementary Table 8). Moreover, the fact that this is reflected in several fitness measure strengthens the evidence that Susu-DRB*13 plays a mechanistic role

(Fig 4; Supplementary Table 9-12). We think the current manuscript highlights this more clearly and we transparently discuss our limitations (e.g., L239).

Moreover, we have expanded our analysis comparing MHC allele and neutral genetic evolution (see response to reviewer 1). While not specific to the alleles Susu-DRB*13, this analysis shows that the evolution at the MHC is stronger than at neutral genetic markers. As per original manuscript, we account for individual expected heterozygosity in all models.

2) Story of the DRB*13 fits the narrative nicely, but what about other alleles? What about alleles that always had high frequency and still seemed to be quite protective, e.g., DRB*1 – TB does not adapt to it? What about DRB*16 – it shows a reverse trend compared to DRB*13 – changes from being somewhat protective to susceptibility. This dynamic would be expected for an allele that used to be rare, rose in frequency and subsequently lost the selective advantage. Is it the case? Finally, what about the frequency of haplotype C given its adverse association with TB – it should drop in frequency? It is hard to gauge the changes in frequency from (a very pretty but not so easy to read) Fig 2D. Also, the sole allele of that haplotype is DRB*4 – but it was omitted from specific tests and graphic presentations. In the supplement (line 60), I found information that this allele was among the few that had “a premature stop codon at amino-acid position 82 and were thus presumed non-functional and excluded from subsequent analyses”. I understand the logic behind this approach, but given what is seen and described for the haplotype C – a re-analysis would be warranted (perhaps for all non-functional MHC alleles) – as a non-functional MHC allele may indeed be a “bad news” during infection... At the same time, for a similar case of haplotype A containing only a non-functional DRB*2 - no such adverse effect is visible (based on Fig 2F).

Authors response: We thank the reviewer for their curiosity about the other alleles and their potential role (see L148; L171; L201), and we apologize for possibly causing confusion as to why specific alleles/STs or haplotypes were or weren't included. Generally, we emphasize in the new version that alleles/STs/haplotypes need to be frequent enough and present in high enough numbers to be considered in further analysis (L126; L382-386). For instance, Allele DRB*16, which the reviewer mentioned, was excluded from further analyses based on the scarcity of the allele in some study years (present in less than 10 individuals) and the

statistical power issues associated with these extremely low numbers. DRB*16 clusters with other alleles in Supertype 4, which was included in analyses and for which the results are reported (see lines 151). As the most frequent allele in Supertype 4, the effect of this allele is likely large, and, indeed, the reviewer is right that the ST shows the opposite pattern compared to DRB*13 (Supplementary Fig. 3).

Frequency changes as observed in the original Figure 2C/D (now Fig 2C) were not an inclusion criterion, but functionality was and is. With the help of Dr. Luke Silver (now coauthor) we annotated the genomic region of the MHC in meerkats and confirmed that our functional alleles fall into this region whereas the presumed non-functional alleles with a premature stop codon fall on a fragment and are likely pseudogenes (L109-112; Supplementary Results). We agree with the reviewer, that it would be great to understand whether such non-functional alleles are bad news, this is a matter for future work looking at these alleles more mechanistically. We redrafted manuscript and no longer include non-functional alleles or supertypes or haplotypes that contain them (see Supplementary Table 6 + 7).

Other, major points:

1) Calculation of frequencies of alleles (and haplotypes as well) – based on what is shown on Fig. 2C (and 2D) it seems that these are not really frequencies – as the values sum to more than 1. What is presented most likely represents “prevalence” of alleles, rather than frequency – i.e., proportion of individual that had given allele/haplotype? Given that the MHC genotyping methodology does not seem to allow assignment of alleles to loci (and so the Authors are dealing with a set of co-amplified variants per individual), looking at the prevalence and associating it with TB is one way to go – but it should be clearly explained and described on the figures. However, whether or not this measure can be “functionally” equated with frequency (as assumed in NFDS) – is another question entirely. For example, from the “adapt to evade” point of view, it can be a very different situation for a pathogen to have to deal with an MHC allele present in half of the population (prevalence = 0.5) but accompanied by several other alleles within each of the individuals, compared to a single locus situation. The notion that after several years of TB presence (and likely adaptation of

this pathogen) higher individual MHC diversity started to translate to a decreasing TB susceptibility may hint to the significance of this distinction.

Authors response: We thank the reviewer for their suggestion. We report relative abundance of the respective MHC measures (Allele, Supertype, Haplotype) within the population, and changed our wording and Figures accordingly (see e.g. Figure 2) but for ease of understanding we refer to the change in the relative allele abundance as frequency (L113).

2) As signaled earlier - the study is purely correlational, and would be much stronger with some functional/experimental data to back it up. I am aware that adding more evidence (for example, data on a reciprocal TB genotype change, any experimental data on infection with the latest TB strain in animals with/without DRB*13, or at least some in silico modelling supporting superior binding of TB antigens to DRB*13)– is unlikely – but I feel it is a weakness that must be recognized, especially when aiming at a journal with such a broad impact as Nature Ecology & Evolution.

Authors response: The reviewer is correct that any of the cited additional evidence would further consolidate our claims, but we do not have this additional data, nor do we think infecting wildlife with an ultimately lethal disease is ethical. We do not think the type of data should exclude ecology or wildlife studies from being published in Nature Ecology and Evolution. Many valuable contributions to Nature Ecology and Evolution are correlational (e.g., Hoelzel et al. 2024, McGregor et al. 2019, Fackelmann et al. 2023, Wisocki et al. 2019, Björk et al. 2022), which later enables testing of specific hypotheses. The quality of the science and rational should be the main criteria. We acknowledge the correlational nature and the lack of reciprocal TB genotype evolution in our discussion (L274, L84).

3) The long latency period of TB gives animals plenty of time to reproduce and pass on genes before eventual death – this fact was also noted by the Authors (l. 294-296). While

they discuss this effect in relationship to dominant pairs, no information on whether TB affected (lifetime) reproductive success was provided – and this would be needed for a full picture of the TB effect on fitness. In general, a proof that TB enforces significant selective pressure to cause allele frequency changes over the course of the study is lacking. Even the number of generations or generation time was not provided for the species. Given availability of genealogies and detailed life histories, calculation of selection coefficients or similar metrics combined with appropriate simulations would strengthen presented findings.

Authors response: This is an excellent suggestion. We have added lifetime reproductive success as additional fitness measure (see responses to reviewer 1). Furthermore, information on the study population, meerkat ecology and life history were added (e.g., Supplementary Fig 1, L85-95; L258-272).

Moreover, based on the reviewers suggestion we included Figure 6 in which we show that the selective pressure applied by TB enforces frequency change from one year to the next in candidate alleles.

4) The specific social & reproductive system of meerkats (which could be described in more detail), with social groups (of related/unrelated individuals?) and dominant pairs might also alter how the overall allele frequencies change in response to pathogen pressure (compared to what is expected under standard population genetics scenarios) – and this aspect should be discussed more directly.

Authors response: We added information to this in the introduction, discussion and supplementary material (e.g., Supplementary Fig 1).

5) Can the length of the latent phase in TB vary? If so, that might be a better feature to study the resilience of individuals (akin to elite controllers in HIV/AIDS studies). I suppose that information was difficult to obtain – but might be more suitable for detection of protective MHC alleles. This aspect could be at least discussed in the article.

Authors response: The time between exposure and clinical signs of infection can vary markedly, lasting on average about 1.5 years, but up to several years have been recorded in

the population (Müller-Klein et al. 2022). Similarly, the time between exposure and infection as well as infection and clinical signs can vary markedly between individuals (see Donadio et al. 2022). Already in the original manuscript resilience was computed and we had evaluated what affected whether an individual survived for longer with symptoms.

As with HIV, TB is cryptic during the latent phase, we do not have information on exact infection times for the study individuals. While exposure has been estimated, assessing e.g. the time from exposure to TB signs does not answer the question whether individuals are more/less susceptible to TB (i.e. do not become infected despite exposure), or susceptible but more/less resistant (i.e. they are infected, but manage to control the infection, leading to a longer latent period), so the interpretation of effects on the approximated latent TB phase will be difficult and the duration might introduce systematic errors between individuals if both processes (susceptibility and resistance) are working simultaneously in the population. However, we already analyze TB progression, i.e. the transition from no clinical TB signs to clinical signs within 3 months, which is a proxy of latency/ capacity to withstand clinical TB infections on a shorter timescale. The results of these analyses are already provided and discussed in the manuscript (see L154; Supplementary Table 9).

6) Measures of TB association with MHC alleles frequencies in a given year/year of death - given long latency, and presumably the retention of a TB strain throughout infection – would not the relationship with frequency of alleles at the time of infection be more informative? Again, I understand how this is unknown – but at present, the confounding factors include not only the time lag needed for parasites to adapt, but also the time lag between infection and actual development of observable symptoms – which further blurs relationship between alleles, their frequencies and susceptibility/resilience.

Author responses: All analyses on TB susceptibility, resilience, progression and mortality are run on individual level, i.e. on the presence/ absence of specific alleles and MHC diversity within the respective individual, controlling for time via year of death (susceptibility/ resilience) or year (progression/ mortality).

The issue of missing information on exact infection status – which would be needed to test the respective allele relative abundance at time of infection – is a shortcoming and no

straightforward test to easily and noninvasively test for TB in wildlife has been implemented within the project. We acknowledge this in L266.

As clinically infected individuals contribute most to TB transmission and clinical infection rather than latent infections are major drivers of mortality, i.e. a selective force, we are convinced the analyses on clinical infection level are still informative and relevant for our specific host-pathogen system.

Minor comments:

- MHC expression – all genotyping is based on DNA analysis, are all (presumably functional = without premature stop codon) alleles expressed by the meerkats? While expression data for MHC class II is harder to get (as it requires preservation of tissues containing MHC II-expressing cells, like APCs, in an RNA-protective buffer), a smaller study showing overall pattern of expression would be useful. Given prominent birth-and-death dynamic within MHC locus, some alleles identified by amplification of exon fragments may really be non-functional, despite an ORF.

Authors response: We thank the reviewer for their suggestion and are delighted to report that we have information on whether alleles are expressed. This was achieved in a collaboration with Dr. Luke Silver and the results are reported in the main text (L109) and supplementary material. The great news are that all our alleles could be located and assigned to the DRB region in the meerkat genome and are expressed.

- Author argue that “our findings of a potential heterozygote advantage in later study years arose from the increased likelihood of carrying Susu-DRB*13 rather than MHC allelic and functional diversity per se.” – is the relationship with high number of alleles/STs maintained when carriers of DRB*13 allele are entirely excluded from an analysis? After all, both mechanisms can act in concert and are not mutually exclusive.

Authors response: We thank the reviewer for their suggestion and we followed suit. We repeated the analyses after excluding individuals carrying allele DRB*13, which indeed reveals that the effects of MHC diversity are not retained (Supplementary Table 13). This

result further indicates that the specific allele rather than overall MHC diversity that is driving the pattern (L180-197, L273-283).

- Line 224: "in individuals" twice

Authors response: This was changed.

Reviewer #3 (Remarks to the Author):

Summary:

By monitoring a total of 3,420 meerkats from the Kalahari, South Africa across a twenty year time span, the authors present an impressive dataset of tuberculosis infection dynamics (including onset of symptoms, progression, and eventually mortality), environmental covariates of disease (social and abiotic), and genetic variation. In particular, the paper examines both neutral genetic data from microsatellites and nucleotide variation in exon 2 of the MHC type 2 gene MHC-DRB. The goal of the study was to test for associations between MHC genetic diversity and disease, with the prediction being that coevolution would produce a signature of negative frequency dependent selection in host immune genes. The authors note that this mechanism of balancing selection has only rarely been observed in nature, likely due to the long span of time that is necessary to observe allele frequency and infection oscillations. Although the paper offers a number of interesting results, the main argument is that shifts in the frequency of a particular MHC allele (DRB*13), as well as the inferred haploypye containing this allele (haploypye F), are causally associated with shifts in tuberculosis best explained by negative frequency dependent selection.

Authors response: We thank the reviewer 3 (Prof. Jesse Webber) for their positive summary of our work, and we are convinced that the updated version further underscores the value of our work.

Major comments:

The data, methods, and much of the main text are strong. This is indeed the longest term, largest (i.e. sample size), and most genetically detailed dataset that I've encountered related

to the evolution of immunity in a natural vertebrate system. Many aspects of this study would be of interest to a large audience of biologists. I particularly commend the authors for their transparency in providing copious summaries of all the important data.

Authors response: Thanks for the appreciation of our dataset and presentation of the results. We are glad to continue a fully transparent presentation of the data.

However, the authors primarily argue that negative frequency dependence drives the observed trends in MHC and disease variation. There are weaknesses to this interpretation that are not well addressed in the current manuscript. These limitations cast doubt on whether this is indeed a strong example of balancing selection, an argument that dominates the introduction and discussion sections of the manuscript. Below, I highlight my major concerns. When possible, I try to identify additional data, analyses, or arguments that might ameliorate these concerns. However, I'll emphasize from the start that the best way to address most of my critiques might be to simply reduce the manuscript's emphasis on balancing selection. The topic could still be addressed with less certainty, and this would open space for increased discussion of environmental correlates with disease, incorporation of some kind of pedigree or spatial information, and opportunities to reflect on both the strengths and weaknesses of the current dataset, including what kinds of data other researchers might want to collect before embarking on similarly ambitious projects.

Authors response: We are delighted with the reviewer's constructive feedback throughout and it inspired us to go above and beyond to improve the updated version.

1. Specifically, we have rewritten parts of the introduction and discussion, reducing the emphasis on NFDS as the driver of the MHC-TB-dynamic. In fact, we re-focused our work more on whether we can disentangle the three non-mutually exclusive mechanism rather than proposing that we definitely are dealing with NFDS. This offers a more nuanced picture than presented in our original manuscript (L100; L179 and following; L273-294; L422 and following).

2. Moreover, we feature ecological and behavioural information more throughout the manuscript (e.g., L85-L97, L174-178, supplementary material) but particularly in the discussion (L258-L272).

3. Finally, we report on limitations in the discussion (also requested by reviewer 1 and 2) in the spirit of offering grounds to reflect on designs for long-term studies, which we think is a great suggestion by the reviewers (e.g., L218, L238).

Concerns:

1. The essence of the negative frequency dependent selection argument is that both host and pathogen populations are coevolving. Lines 75-78 argue that “robust evidence” only requires 3 types of data. The second type of data (b: strong pathogen selection) misses two additional requirements: showing that the pathogen population itself has evolved, and proving a causal connection between host and parasite alleles whose frequencies negatively covary over time. This study offers no data to support pathogen evolution or direct functional connections between host alleles infections resistance/susceptibility. Moreover, of the 4 important transitional states noted in Fig 1D, at best this paper only captures states 2 and 3. These limitations should be clearly stated. If there happen to be blood or tissue samples from the pathogen that could be used for genetic analyses, this would be an extremely important addition to the study that might be able to fill the pathogen side of this evidence gap.

Authors response: Reviewer 2 also suggested the lack of information on pathogen evolution as a shortcoming and we agree. This data is however not available to us and we communicated it as a drawback in the discussion (L283-286).

2. In line with criticism 1, and with the author’s own criteria (Lines 75-78), it’s important to identify all possible reasons why host allele frequency oscillations might occur even in the absence of pathogen evolution. Although statistical analyses (Fig 2 E,F and Fig 3) do show significant associations between DRB*13 (or Haplotype F) and TB that change in direction over time, the trends are weak and could be biased by infection dynamics within particular

social groups. For example, DRB*13 and HapF start at low frequencies and both drop to very low levels between 2002 and 2008. There really does need to be some discussion of how this rarity could influence false positive associations. For example, if those alleles reside in a social cluster that got hit hard by TB during those years then it would show an elevated signal of susceptibility driven more by rareness across social groups and stochasticity in infection prevalence than a functional connection to disease. I looked pretty carefully at the various statistical models and think the following would be important covariates to add:

-Social cluster ID

-Prevalence of infection (i.e., frequency of individuals with signs of infection, not just the presence/absence of infection in a group)

Authors response: We thank the reviewer for their suggestions. While we disagree, with the reviewer that the allele effects are weak (in the model results they count among the stronger impacts), we considered these suggestions carefully.

Of course, meerkat social structure has a strong influence (seen already as the within group discrepancies between dominant and subordinate meerkats), and ideally the analyses could be run on group level. However, based on the high number of individuals and sparse group representation, including social group led to overdispersion and violation of model assumptions, so we decided to run the analyses on the population level. As male emigration and immigration into other groups and evictions of females happen regularly, which then often attempt to establish new groups, there should be considerable gene flow between groups, which is also why each group cannot be seen as a population (similar to the hefts in the Soay sheep). Future work might be able to disentangle the network and group formation over time and its consequences for genetic diversity but this is beyond what we can do with our data.

We like the suggestion of including prevalence of infection. The issue with this variable is that it does not surmount to TB risk. Due to social hierarchies, high and low risk behaviours and the considerable uncertainty around TB latency, we think the variable might poorly reflect TB risk of the individual, while it reflects TB prevalence at the population level.

3. Connected to criticism 2, the discussion section highlights how the dominance system of

meerkats might make them particularly susceptible to TB. I was intrigued by lines 294-299. If dominant individuals don't turnover in populations then it doesn't seem like there's much opportunity for the population to evolve new MHC alleles. This begs the question, how much evolution really could occur during the timespan of the study, and how exactly did the observed changes in DRB*13 take place? Given that the authors have data on group size over time, as well as some pedigree information on individuals (line 326 notes that relatedness was calculated for haplotype calling), it would be helpful to identify which social groups are driving the late increases in the DRB*13 allele, and describe what happened to those groups that allowed for evolution during the study. I'm particularly concerned that the trend of increased DRB*13 could come from a small number of groups with very low exposure rates late in the study.

Authors response: We thank the reviewer for their suggestion. We assessed this by looking at the emergence of TB in groups for every 3 months period, accounting for the presence/absence of DRB*13 in the group at the respective period. The results suggest there is no significant difference in TB prevalence between groups with Susu-DRB*13 and without (Wilcoxon test=180325, $p=0.369$, see Figure below; N group periods: Without DRB*13: 1040, with DRB*13: 439).

4. In lines 144-155 the authors highlight TB associations that may be separate or jointly connected to effects of allele DRB*13 (or the related haplotype). The most relevant of these is the MHC diversity data: individuals with greater DRB diversity had greater TB resistance at the end of the study. The authors provide some data to suggest that this trend is driven by DRB*13 being present at high rates in MHC diverse individuals. These arguments are not persuasive:

a. Haplotype F happens to be one of a few that contains three distinct alleles. DRB*13 would thus necessarily be dragged along if selection was simply for diversity. This is not discussed in the paper.

Author response: This is a very good observation and was also noted by the other reviewers previously. To account for this, we repeated other analyses after excluding individuals carrying DRB*13, resulting in a loss of the MHC diversity effect, which indicates that DRB*13, rather than MHC diversity, is the main driver of our results on TB. But, as we cannot be sure we discuss it (L273-283)

b. Given the size of the datasets and the number of models tested, some effort should be made to compare the model and, ideally, identify which one best explains the infection data. There should at least be a description of R^2 values of models focusing only on DRB*13 or only on allelic diversity/heterozygosity. Another option might be to apply some kind of SEM/path analysis. There may not be enough breakdown in covariance between DRB*13 and allelic diversity to inform causal influences, but simply showing that there's a greater proportion of high allelic diversity individuals with DRB*13 than expected by chance is not compelling. If the authors choose to keep this argument, at very least the statistics need to be restricted to only those social groups where there was even a chance of inheriting the relatively rare DRB*13 allele.

Authors response: We report the R^2 from linear modelling here for the reviewer. Models including the allele scaled consistently higher compared to the models including only Number of DRB-Alleles. For the survival models we were unable to find a neat way to compare them, which is why we supplied the AIC and BIC results. Does the reviewer still think this is worth including in the main text?

Model	Susceptibility	Resilience	Progression		Survival	
Nr alleles model	0.083	0.143	AIC	BIC	AIC	BIC
			267.45	199.77	1492.04	1359.29
DRB*13 model	0.112	0.172	AIC	BIC	AIC	BIC
			272.99	193.37	1491.97	1332.68

c. Figure 6 builds on arguments that are not completely in line with negative frequency dependent selection, at least not across cycles of negative vs positive selection. It all depends on which stage in Fig 1D the process is occurring. During some points in time there should be a positive association between a causal allele and infection, and at others there should be no or negative correlations. The allele freq data for DRB*13 suggest that the study is observing stages 2-3 in Fig 1D. This distinction is important when making predictions about positive or negative correlations within vs between generations (Lines 205-216). The authors also don't provide much effort toward explaining the disconnect between looking at allele frequencies vs haplotype frequencies. Even with a smaller sample size there should still be some trace of an effect, unless the effect size is quite small.

Author response: Given your feedback and that of other reviewers we opted to remove this figure and analysis. But the reviewer is right that we interpreted our results with the specific phase of the cycle in mind, as pointed out in your comment, but did not make this explicit. Following an earlier suggestion from another reviewer we instead now show change in frequency between years and in relation to pathogen prevalence (Fig 6). We think this now represents a more interesting view because it implies that at higher TB risk the resistance associated allele Susu-DRB*13 tends to increase in frequency. Together with the other results on survival and LRS, this suggests Susu-DRB*13 might indeed under TB-mediated selection (irrespective of where we are in we are positioned in the negative-frequency dependent cycle).

5. Building on the finding of allelic diversity correlated with infection resistance, the topic of

DRB functional supertypes gets rather short shrift in the main text. As I noted earlier, it's great that the authors include much of the ST data and models in the supp mat. But in Supp Fig 2 it appears that a number of other supertypes switch between positive and negative TB associations through the course of the study. The text only focuses on ST9 (containing DRB*13) and ST4. It is convenient that DRB*13 is in its own supertype, but given that supertype is probably a better predictor of functional similarity between alleles, there is an argument to be made for refocusing the paper on supertypes rather than individual alleles or haplotypes. If the authors choose to stick with haplotypes, additional justification should be provided.

Authors response: Like with the different p-distances, a focus on alleles, supertypes and haplotypes tells slightly different stories. Supertypes do provide a broad overview of immune recognition patterns, but are often resorted to as a way to increase statistical power by grouping alleles. We reported supertypes for completeness, but mainly discuss alleles and haplotypes because our study has sufficient power. They may also at times blur patterns and we see a generally lower association strength in our cooccurrence analysis with regards to supertypes (Supp. Fig 3A). MHC haplotypes specifically account for the specific combination of alleles across the whole set of MHC loci, and since they represent alleles inherited together represent evolutionary units. The expected evolutionary dynamics are therefore also different. Haplotype F, for instance, included the most common Susu-DRB*1 and Sus-DRB*13. A negative effect of one allele could hence balance out the positive of another and vice versa. Being found often in combination with Susu-DRB*13 could, for instance explain, why Susu-DRB*01 was not negatively associated with TB measures, but the common allele does reduce LRS. We recognize this is inferential evidence. We added our reasoning for also looking at haplotypes in the method section on MHC haplotyping (L342).

7. Figures 4 and 5 are discussed only briefly in the results. The connections between TB resilience, life expectancy, disease progression and mortality are extremely interesting but not thoroughly synthesized in the discussion section. Why are each of these metrics important to add as main figures? Is it just in testing whether there's a genetic association with each of them? How do the genetic associations differ between metrics, even if only focusing on DRB*13, and why might this matter for understanding how evolution is affecting

infection dynamics?

Author response: While re-writing the manuscript we kept this in mind and tried to link them more strongly. For instance, the relationship between living longer and reproducing more is now prominently featured in the discussion (e.g., L253).

I greatly appreciate the effort that the authors made in conducting and writing up this interesting study. I hope that they find these comments to be relevant and useful.

-Jesse N. Weber

Author response: We found the comments very helpful and hope we could address them with sufficient clarity and rigor.

We thank all reviewers for their helpful comments and contributions to increase the clarity and rigor of our work. Please see point-by-point responses below.

Reviewer #1 (Remarks to the Author):

I appreciate the authors' efforts in addressing my concerns and suggestions, particularly in the discussion section. While I am satisfied with most of the revisions, I believe there are still some points that need to be addressed before the manuscript meets the standards of Nature Ecology & Evolution. Specifically, the authors should present all results comprehensively and include additional tables detailing sample sizes and phenotype distributions. Below, I outline my remaining concerns in relation to my previous comments:

1) Negative frequency-dependent selection (NFDS), fluctuating selection (FS), or heterozygote advantage (HA)?

The authors have made substantial rewording to better reflect this point. They have also provided a clear explanation of the relationship between DRB*13 and MHC diversity. I am satisfied that they now acknowledge their inability to differentiate between different pathogen-mediated selection (PMS) mechanisms.

Authors response: We are happy that this was addressed sufficiently. We thank the reviewer because their feedback added clarity and improved the scientific rigor of our work.

2) Evidence of selection is not solid without lifetime breeding success (LRS)

I appreciate the addition of models testing the association between MHC variation and LRS. However, I notice that the MHC-LRS association does not appear to be time-dependent or cyclical. This may be due to a lag between TB infection and LRS, which the authors could address in the discussion. Additionally, since they mention reproductive skew, they should provide a supplementary figure illustrating this. If the skew is significant, they could consider applying Bayesian modeling (e.g., MCMCglmm).

*Authors response: We thank the reviewer for their suggestion to include LRS in the very first place. We elaborate on the LRS results now in the result section (L171) and dissect its role further in the last paragraph of the discussion. We also were aware of the reproductive skew as the Supplementary Figure 1 showed. In fact, we had tried Bayesian modelling before with this data but the results did not differ (see excerpt from Supplementary Table 12 below and exemplary Table for Susu-DRB*13 results using the Bayesian approach) and even effect sizes remained similar. Thus, we opted to stay with the frequentist modelling approach for consistency.*

Supplementary Table 12 excerpt: Frequentist approach using generalise linear models as described in the method.

	Estimate	CI.2.5	CI.97.5	p-value
Intercept	2.18	0.907	3,452	0.001
DRB*13	0.992	0.232	1,752	0.011
DRB*13:Year	0.025	-0.401	0.452	0.908
DRB*13:Year[^2]	-2,581	-4,585	-0.577	0.012
NR DRB-Alleles	-0.022	-0.217	0.173	0.823
NR DRB-Alleles:Year				

NR DRB-Alleles:Year[^2]				
NeutralGenDiv	0.694	-0.507	1,894	0.257
Age [Death]	5,064	4,337	5.79	<0.001
Sex [M]	0.259	-0.291	0.809	0.356
Sex [M]:SocStatus [Sub]	-0.731	-1,395	-0.066	0.031
SocStatus [Sub]	-0.96	-1,451	-0.469	<0.001
Year	-0.752	-1,049	-0.455	<0.001
Year[^2]	0.084	-0.541	0.71	0.792

Exemplary Table for Susu-DRB*13 results: Bayesian approach using MCMCglmm with a weak prior, 1000 burnin, 5000 permutations and Poisson distribution (because negative binomial is not supported by MCMCglmm).

	Estimate	CI.2.5	CI.97.5	p-value
Intercept	1.63	0.27	2.77	0.010
DRB*13	0.82	0.12	1.82	0.010
DRB*13:Year	-0.06	-0.70	0.72	0.945
DRB*13:Year[^2]	-2.39	-4.83	-0.12	0.030
NR DRB-Alleles	-0.12	-0.57	0.34	0.595
NR DRB-Alleles:Year				
NR DRB-Alleles:Year[^2]				
NeutralGenDiv	0.28	-0.79	1.52	0.685
Age [Death]	5.88	5.28	6.68	<0.003
Sex [M]	-0.69	-1.25	-0.12	0.010
Sex [M]:SocStatus [Sub]	-0.005	-0.70	0.72	0.031
SocStatus [Sub]	-1.30	-1.25	-0.120	<0.003
Year	-0.90	-1.24	-0.47	<0.003
Year[^2]	0.44	-0.36	1.26	0.340

3) Allele frequency change should be tested using pedigree data

Unfortunately, the authors do not have sufficient pedigree data to perform a gene-drop analysis. However, they have plotted allele frequency changes against TB prevalence over time, which serves as a valuable supplement to their claims.

Authors response: We thank the reviewer for validating our analytical choices.

4) Non-significant statistical results are not reported, and there is no information regarding multiple testing corrections

This section still requires substantial improvement. In the supplementary tables, only some of the MHC alleles shown in Figure 2A (above the 5% threshold) are reported. For instance, results for DRB*06, DRB*10, and DRB*11 are missing. It is essential to provide a complete list of all eligible alleles, as well as corresponding results for supertypes and haplotypes.

Authors response: Thanks for your suggestion but we have specified in the original (L364) and revised manuscript (L384) that we only tested alleles that were sufficiently present in all years and show repeated associations with TB. DRB*06, DRB*10 and DRB*11 do not qualify. This was done to avoid spurious results or MHC-associations across years that are untraceable. We also had specified information regarding multiple testing in all versions including the current one (L360).

5) Caveats of the study are not sufficiently discussed

In response to my previous comment, I believe the authors misunderstood my point about the study potentially being underpowered. While they have conducted thorough quality control to obtain reliable allele data, they have excluded low-frequency alleles (below 5%). Although this is a necessary step in amplicon sequencing to minimize sequencing errors, it also removes potentially novel alleles that could arise from recombination or mutations. Since low-frequency alleles are likely to be positively selected under NFDS, their exclusion may reduce the study's power to detect selection signals.

Authors response: We apologise for having misunderstood the reviewers previous point and thank you for clarifying. We agree that with our approach the emergence of novel (possibly resistant) alleles is unlikely to be uncovered. As we continue to investigate the population, the emergence of novel alleles is an interesting future endeavour, but we think our choices strike a fine balance between evolutionary underpinnings of NFDS and sufficient statistical power to detect NFDS (or other forms of selection on the MHC). Additionally, NFDS can also arise from recycling of already present alleles. We do not want to ignore the reviewer's valid criticism, however, and have therefore remarked upon this potential downside of our choices in the methods (L385).

6) Lack of detailed information about the study system

I appreciate the addition of a sentence further describing the study system in the main text. Furthermore, Figure 2B and Supplementary Figure 1 address my previous concerns. However, I strongly recommend including a supplementary table to provide essential details about the sample size, as this is crucial for evaluating the study. The table should include:

Sample size for each year (total, MHC-genotyped, and TB-observed individuals)

Number of social groups per year

Distribution of key phenotypic traits, such as LRS for males and females separately

Authors response: We thank the reviewer for recognising the added figures and information. We are happy to provide the additional tables (see Supplementary Tables 1 and 2).

Reviewer #2 (Remarks to the Author):

I appreciate the effort the authors have made in thoroughly addressing the reviewers' comments. As the focus has shifted away from NFDS, I now find myself somewhat struggling to identify truly impactful findings. The authors propose a unique feature of their work was an observation of a switch from TB susceptibility to resistance in the MHC allele Susu-DRB*13. However, what remains unclear (even as a hypothesis) is the underlying mechanism driving this shift. This finding is intriguing, given that the allele was throughout the study of rather low abundance (including the time of susceptibility) and only began increasing in frequency once it started conferring resistance—a pattern that aligns with NFDS. Yet, if this allele was rare and associated with susceptibility, one could expect it to be lost from the population rather than to start to confer resistance? Or would the Authors argue that because it was not very common, pathogens “evolved away” and so it could become eventually protective? All is even more complicated, as the most abundant allele, *01 (which often accompanies the *13 on the F haplotype), appears to provide stable

protection, as if pathogens did not adapt to it at all. While the findings are supported by an impressive dataset in a fascinating host-pathogen system, their broader implications remain somewhat limited. The most striking takeaway after all may be that DRB*13 was for a long time rare, but increased in frequency once it became protective. If the study had captured a full cycle of Red Queen dynamics (i.e., high frequency leading to susceptibility, a subsequent decline in frequency, followed by the acquisition of resistance and a resurgence in frequency), that would truly represent the Holy Grail of observational MHC studies. Also, what I find missing from the discussion—perhaps due to space constraints—is a reconciliation of some of somewhat conflicting observations surrounding the focal allele *13 (and its associated supertype or haplotype):

Line 154-155: “TB positive individuals carrying Susu-DRB*13 (and its corresponding ST09 and haplotype F) were more likely to progress from exposure to clinical signs of TB over the course of the study” – how this observation can be reconciled with supposedly protective effect the *13 allele conferred since 2010? They progressed but lived with it, suggesting some sort of control over infection>

Line 160-169: “Finally, individual lifetime reproductive success, as the ultimate Darwinian fitness measure, peaked between 2005 and 2013 in individuals that carried Susu-DRB*13”- it’s interesting, as it seems to coincide with higher risk of TB infection faced by carriers of *13 until early 2010s?

Author response: Thank you for your assessment. The reviewer is correct that the increase in frequency following the gaining of resistance aligns with NFDS and this is a unique finding. Yet, in times when alleles confer susceptibility they may not necessarily be lost (see Soay sheep populations with somewhat stable allele frequencies; Huang et al. 2022), as one generally expects a diverse set of different pathogens driving selection at any given time and disadvantageous alleles against one pathogen may confer resistance to other. Many MHC alleles have therefore long “generation times” and are maintained even beyond species barriers (as is the conceptual underpinning of trans-species polymorphism). In social animals and meerkats with a cooperative breeding system in particular, allele turnover (and with that both decline and increase of alleles following selection) may actually reduce and recycling of alleles that are maintained at low frequency is possible. So, one option is that as the reviewer’s suggests that its DRB*13s low prevalence in the population may have meant selection on the pathogen was reduced and it evolved away from DRB*13. Alternatively, selection by other pathogens (or even other selection pressures) maintained MHC polymorphism and Susu-DRB*13 was kept. As stated in the previous response, we unfortunately do not have information on the pathogen(s) other than whether the host is TB positive or not to confirm this (and we acknowledged this in the discussion).

We apologise there was an error in the L154-155 since we flipped the sign (from positive to negative) but the text was not altered. This was now corrected and is no longer conflicting. The statement in L160-169 is correct as such and we expanded on discussing this in line with suggestions expressed by reviewer 3.

Minor points:

Fig 2 – in caption description of panel E and D are mixed up.

Author response: Thanks. This was corrected.

Fig 3. Maybe I am misunderstanding or there is a mix-up – but I am not sure how there would be a non-linear relationship for Susu-DRB*13 if TB Hazard ratio on the lefthand side

plot of B panel is significant for DRB*13:year but not DRB*13:year²? In contrast, the C panel shows significant effect of DRB*07:year², but the relationship is supposedly linear for DRB*07 (and so is shown on a righthand side of the C panel)?

Author response: We thank the reviewer for their comment. The author is right that the time-dependent TB-DRB*07 association was non-significant after correcting for multiple testing (as can be seen from Supplementary Table 8). We therefore removed the plot to avoid confusion, and leave the result section as it is because the results were reported correctly there (previous version: L149; current version: L151), stating: "...TB susceptibility was generally increased in meerkats carrying Susu-DRB*07 (Estimate: 0.68, CI: 0.25– 1.21, p-value < 0.01, Fig. 3DE),..." – we apologise for the mix up.

Regarding the non-linear relationship for Susu-DRB*13, since the dredge function did not remove the quadratic terms of the complete model, their inclusion in the model means the predicted line can still be non-linear and reflects the combination of these terms.

Reviewer #3 (Remarks to the Author):

I thank the authors for their thoughtful and thorough response to my original critique. The largely rewritten manuscript provides important new data/insights (i.e., estimates of lifetime reproductive success, evidence that MHC alleles are expressed, neutral vs MHC genetic divergence, and more explanations of socio-environmental influences). The new organizing theme (disentangling the three causes of balancing selection) is also nuanced and more informative than the original focus. Despite the lack of a causal connection between MHC and disease, the paper still offers an impressive breadth of data and insightful analyses and discussion that would be of interest to NEE readers.

Authors response: We thank Prof. Jesse Weber for his feedback. Similar to the previous round we found it most constructive and thought provoking.

Although the paper has improved markedly, I still have a few concerns.

Major comments

1. The term "co-evolutionary arms race" in the title is not one of the main results from the paper and the title should be rewritten. While pathogen-host coevolution is important for maintaining MHC diversity, nothing in this article documents TB evolution. Any sentences suggesting evidence of coevolution (e.g., lines 38,214) should also be rephrased. That said, the authors do a much better job of noting throughout the manuscript that multiple selective forces can drive the apparent shifts in MHC allele frequency.

Authors response: Thank you for your suggestion. We followed suit. The new title now reads: „Twenty-years of Tuberculosis-driven selection on meerkat MHC". Other references to co-evolution were removed (e.g., L38, L205 and L216).

2. Comment #7 in my original review (biological story explaining patterns of selection) could still be more thoroughly addressed. I agree that lines 253-257 tell some of the story, and the final paragraph nicely restates the goal of excluding some types of selection. But I'm struggling to think of a straightforward explanation for how selection (either rare allele advantage or divergent selection) could explain observed changes with DRB*13. Here's an example of how my logic keeps falling apart:

a. Given: DRB*13 correlates with increased susceptibility and resilience from 1999-2013.

After that time, susceptibility decreases for the allele while resilience remains high. Also, reproductive fitness for individuals with the allele peak during the early period, but then drops to average levels for the whole population.

b. If the DRB*13 allele itself didn't change in 2013 (i.e., no mutation/functional shift), then what could cause a decrease in susceptibility? This is where I'd really like to hear more biological predictions from the authors. I don't think this shift in susceptibility could plausibly be explained by a rare-allele effect over such a short time span. A bit of evidence is given against heterozygote advantage and fluctuating selection, but there isn't much critique of rare alleles driving this specific kind of pattern.

c. One could posit that TB has evolved. But there doesn't seem to be a clear reason for the disease to evolve at this point. It is already increasing in population size by targeting/infecting the expanding number of DRB*13 individuals. Shouldn't this increase TBs spreading potential given that those infected individuals are living longer.

d. Are there any general environmental/social changes that occur around the time of the susceptibility change? These are discussed as factors in the model, but I don't remember them being tied to any particular years/time frames.

e. The only reasonable scenario I came up with is that DRB*13 containing individuals with clinical signs might be less likely to pass on the disease, leading to decreased prevalence (not necessarily susceptibility) in progeny of the same social group that contained the DRB*13 allele in later years.

f. The most important point here is that the paper mainly ends with generalities that may cause selection variation (lines 258-294). I applaud the authors for embracing the very difficult challenge of discriminating between selective forces. But, I don't see how any of the listed explanations should drive the clear directional shift in DRB*13-TB susceptibility around 2013. I fully admit that this confusion may derive from my ignorance about the study system. A general reader would likely find this hypothetical context useful, and I couldn't help but challenge the authors to think of a few more possible explanations.

Author response: Thank you for outlining your thoughts. Its indeed a rather complex dataset and as the reviewer noted and as we pointed out (L266 previously and now in the last paragraph of the discussion) environmental, social and genetic factors are at play at the same time as TB. It makes perfect sense that the reviewer would like a fully disentangled story and we do not want to end with generalities, so we have tried to merge meerkat social dynamics, environmental pressure and evolutionary concepts in the last paragraph of the discussion. This is in part based on our data and inference using other findings about this populations.

For clarity we shifted what was the last paragraph to be the penultimate one, and move the former penultimate paragraph last and add contextualising data and references. Due to space constraints and to minimise confusion, we largely focus on meerkats (rather than also talking about badger or buffalos), trying to construct a cohesive and evolutionarily sound story.

3. I was confused by some of the word choice in the discussion. The authors may conflate the terms resistance and resilience at several places. I don't think it's primarily "resistance" (presence/absence of infection) that's associated with the fitness benefit: lifetime reproductive success increases before 2013 while susceptibility is also increasing), but rather constantly highly resilience (lines 245-246). Granted, decreased susceptibility may contribute more in later generation. This distinction is important because some of the

citations for MHC are specifically connected to resistance. Part of the problem here may be how I (and many others— See Bishop 2012 10.3389/fgene.2012.00168) think about pathogen resistance. This Bishop paper really does relate well to this work: could DRB*13 actively change selection on the pathogen (in which case it is resistance)? If so, at what time points? If not, then this is closer to tolerance. With the low levels of TB infection in meerkats, Bishop suggests that tolerance is difficult to understand theoretically, let alone measure.

Author response: We appreciate the reviewer's suggestion and have read the Bishop 2012 paper and re-read Raberg et al. 2007 and 2009, which we are more familiar with. Both resistance (as a measure of infection intensity/parasite load), and tolerance as the rate of change in fitness as infection intensity/parasite load increases (Raberg et al. 2009), require quantifiable infection intensity/parasite load. However, not having TB or having TB is binary. The reviewer is right that we need to be careful and we thus removed reference to resistance from our results (e.g., see L103, L216, L253, L294).

But to answer the reviewers intriguing question of whether DRB*13 changes selection on the *M. suricattae* and at what time, we can only speculate (and thus do not discuss this point in the discussion) that the increase TB prevalence in the late 2000s and peaks in early to mid-2010s possibly meant sufficiently strong selection on Meerkats to favour individuals with certain beneficial variants (e.g., Susu-DRB*13). As a consequence of the variants increase the bacterium *M. suricattae* may have experience selection in the latter half of the 2010s and this is also where we see a subsiding of general exposure (even though TB symptoms are still common due to the time lag of the diseases). It would be most interesting to look at the divergence in *M. suricattae* at those critical points in time. Regarding tolerance, as Bishop suggests and we know from Raberg tolerance is a difficult concept, which we avoided, because we do not have quantifiable infection intensity.

4. While I like the new framework for the paper, Figure 1B may benefit from some editing. The allele freq distribution and population structure rows are nicely suited to this study, but the other two columns are a bit of a stretch. Pathogen load is presented as a quantitative trait, while here it is binary. And this study doesn't address pathogen diversity. Perhaps this figure could also be used to highlight which of the patterns are tested in this paper, and which are supported?

Author response: We understand the criticism regarding Figure 1B. We had included expectations for MHC associations with pathogen diversity for completeness because Richardson and Spurgin (2010) outlined specific predictions for this type of analysis. The reviewer is right in that we do not have this type of data. We have removed that row and briefly touch upon this in the discussion (L288). Additionally, we refer to infection likelihood now instead of infection load in Fig 1B. We also like the reviewer's suggestion to pinpoint which patterns are supported by our results and indicated in Figure 2-6, which mechanism of PMBS is supported by each result.

5. The idea of connecting TB prevalence to change in relative prevalence of the DRB*13 allele is very good, but the statistics underlying Fig6B were unclear (text in lines 198-200, 434-436). The lagged approach that the authors are referencing asks if prevalence in the previous year predicts the amount of change in the next year. This is the most conservative approach, and the one I would favor. Why is the approach in this manuscript "similar to" (lines 434-436) and not just a "lagged approach"? Some of the methods text led me to think

that the authors are testing whether prevalence in the current year is a predictor of change in allele prevalence from the previous to current year, or of changes within the current year.
Author response: Thanks for confirming that the lagged approach is a good one and we apologise for the confusion about how this was done. We used the lagged approach. With the statement we wanted to express that this approach was used previously in a vole population. In other words, we meant that the analytical approach is the same but the systems are only similar in that their and our study are host MHC-pathogen association studies. To be clear, we did not model the frequency changes within year, but between one year and the next as per “lagged approach”. We amended the text to reflect that (L205 and methods).

6. Because this is a correlative study, please try to remove causal statements about the results (i.e., line 102 “determines” should be “correlates with”).

Author response: Thank you. We changed both instances where we used causal statements.

Minor comments:

1. Line 163: independent of “time”, perhaps change to “year”.

Author response: Thanks. We exchanged time for year.

2. Line 250: I previously asked about the R^2 value for the models, and I appreciate the authors providing that info in their response. I don't necessarily think that the R^2 needs to be reported in this paper. But I can't help but revisit my previous effect size point. When citations like this one are made (single allele ~ 48% of variation SIV survival—a huge effect!), it begs the question of just how much of the variation in this study is explained by DRB*13. The model fits in the author's response letter (~11% and 17% for susceptibility and resilience, respectively) include both the DRB*13 allele and a few other factors, suggesting that the allelic correlations (or at least MHC correlations) are small relative to some other host-pathogen studies.

Author response: The reviewer brings up an interesting point and we agree generally. First, we want to highlight that the cited study was experimental and thus does not recapitulate natural selection conditions. Secondly, the random effect structure of the coxme prevents a traditional model comparison with variance explained, and only pseudo- R^2 may be calculated (but this is not equivalent of variance explained). We agree that leaving this statement about SIV survival in macaques as is would beg the question about the effect of MHC on meerkat survival. We therefore added as sentence in L260 that the effect of the MHC is subtler in our wild population. We arrive at this conclusion looking at effect sizes of individual variables.

3. Lines 425-426: grammar

Authors response: Thanks this was corrected.

4. Line 420: “were” -> where

Authors response: Done.

5. Line 452: Colored boxes represent alleles...

Author response: Thanks. We changed the text accordingly.

6. Fig 2 legend: captions D and E out of order.

Author response: Thank you. This was addressed.

As with the original submission, I hope the authors find these comments to be relevant and useful. This is important and well-done research. I appreciated the opportunity to read and learn from this manuscript.

-Jesse N. Weber

We thank all reviewers for their feedback. We have tried our best to include the requested changes and carefully amended the manuscript to reflect both strengths and weaknesses.

Response to Reviewers:

We thank both reviewers for their helpful comments throughout the peer-reviewing process and are happy that we are able to fully comply to the last request by reviewer 1. We amended the supplementary table as per their request (see below).

Reviewer #1:

Remarks to the Author:

Thank you very much for the revised version and I believe it addresses my concerns well. I am looking forward to seeing it published soon.

A minor suggestion: I recommend updating the text in Supplementary Tables 1 and 2 for clarity. Perhaps use "complete dataset" and "MHC-typed dataset" instead of "total" and "our subset" to avoid potential confusion.

Reviewer #3:

Remarks to the Author:

I thank the authors for carefully addressing the reviewers' comments. I have no further concerns about the content of this manuscript. It represents an impressive and important project that will be of great interest to readers of NEE.